# A large Canadian cohort provides insights into the genetic architecture of human hair colour

Frida Lona-Durazo [1✉], Marla Mendes[1,2], Rohit Thakur[3,4], Karen Funderburk[3], Tongwu Zhang [3,4], Michael A. Kovacs[3], Jiyeon Choi [3], Kevin M. Brown [3] & Esteban J. Parra [1✉]

Hair colour is a polygenic phenotype that results from differences in the amount and ratio of melanins located in the hair bulb. Genome-wide association studies (GWAS) have identified many loci involved in the pigmentation pathway affecting hair colour. However, most of the associated loci overlap non-protein coding regions and many of the molecular mechanisms underlying pigmentation variation are still not understood. Here, we conduct GWAS meta-analyses of hair colour in a Canadian cohort of 12,741 individuals of European ancestry. By performing fine-mapping analyses we identify candidate causal variants in pigmentation loci associated with blonde, red and brown hair colour. Additionally, we observe colocalization of several GWAS hits with expression and methylation quantitative trait loci (QTLs) of cultured melanocytes. Finally, transcriptome-wide association studies (TWAS) further nominate the expression of *EDNRB* and *CDK10* as significantly associated with hair colour. Our results provide insights on the mechanisms regulating pigmentation biology in humans.

[1] Department of Anthropology, University of Toronto at Mississauga, Mississauga, Ontario, Canada. [2] Departamento de Genética, Ecologia e Evolução, Instituto de Ciências Biológicas, Universidade Federal de Minas Gerais, Belo Horizonte, MG 31270-901, Brazil. [3] Laboratory of Translational Genomics, Division of Cancer Epidemiology and Genetics, National Cancer Institute, National Institutes of Health, Bethesda, Maryland, USA. [4] Integrative Tumor Epidemiology Branch, Division of Cancer Epidemiology and Genetics, National Cancer Institute, National Institutes of Health, Bethesda, Maryland, USA. ✉email: frida.lonadurazo@mail.utoronto.ca; esteban.parra@utoronto.ca

Pigmentary traits in humans (hair, eye and skin pigmentation) have a polygenic architecture, in which pleiotropy and epistasis are common phenomena[1–5]. Contrary to other complex traits, the environment has little or no effect on pigmentary traits, with the exception of facultative skin pigmentation (i.e. tanning ability)[6,7]. Therefore, the wide range of diversity in pigmentary traits across worldwide populations is mainly due to genetic variation that has been shaped by a range of evolutionary factors (i.e., drift, gene flow and selection)[8–12]. Elucidating the genetic architecture of pigmentary traits may aid in the general understanding of biological pathways, gene interactions, their regulation and expression. Likewise, it has the potential to further identify molecular mechanisms associated to important diseases, such as skin cancer (e.g., basal cell carcinoma, squamous cell carcinoma and melanoma).

Current evidence indicates that there is a partial overlap in the genetic architecture of hair, eye and skin pigmentation. Some genes (e.g., *OCA2, TYR, SLC24A5*) have been associated with multiple pigmentary phenotypes, whereas other genes have been associated with only one of these traits (e.g., *MFSD12*- skin pigmentation)[13]. Adding to this complexity, previous studies have identified the presence of allelic heterogeneity, in which different variants within a single gene are associated with pigmentation variation in different populations (e.g., *MFSD12, OCA2*)[5,13,14], as well as the effect of multiple independent variants on the same locus associated with a diverse range of pigmentation tones within populations (e.g., *HERC2/OCA2, MC1R, GRM5/ TYR*)[5,15–19]. Furthermore, some genes associated with pigmentary phenotypes (e.g., *ASIP, MC1R, TYR, SLC45A2, OCA2, IRF4, SLC24A4*) are also known to increase the risk of cutaneous melanoma[20–26]. Thus, the link between pigmentation and cancer risk also highlights the biomedical importance of efforts to characterise the genetic architecture of pigmentary phenotypes[27].

Hair colour is a quantitative phenotype that results from differences in the amount and ratio of eumelanin and pheomelanin synthesised in melanocytes located in the hair bulb, which then migrate to the hair shaft[28–31]. Recently, large-scale genome-wide association studies (GWAS) have uncovered numerous loci associated with hair colour in people of European ancestry by using discrete hair colour categories as an approximation[32,33]. By analyzing the UK Biobank (UKBB) data, both of these studies identified hundreds of regions associated with the phenotype in question across the genome, some of which had not been previously identified, due to the lack of power to detect significant associations (e.g., *TSPAN10, FRMD5*).

Furthermore, Morgan and colleagues provided a detailed analysis of the loci associated with red hair colour, including penetrance and interactions among single-nucleotide polymorphisms (SNPs), offering insights on the genetic complexity of this hair colour tone[33]. In spite of these advances, for most of the hair pigmentation-associated loci, the causal variants and the molecular mechanisms underlying pigmentation variation remain to be identified[34]. This is in fact a challenging task, given that most of the top SNPs associated with hair colour are located in nonprotein-coding regions of the genome, with no obvious or direct function on the trait, thus hinting to a regulatory function.

Major advances in the characterisation of regulatory elements, along with the development of a diverse set of computational tools using GWAS summary statistics, have facilitated the interpretation of GWAS hits of several polygenic phenotypes[34,35]. For instance, statistical fine-mapping methods have become computationally feasible and make it possible to model multiple causal variants simultaneously[36]. Additionally, when putative causal variants are identified, they can be further explored by evaluating their effect on the regulatory profile of target genes on relevant cell types, which can be statistically tested with colocalization or transcriptome-wide association studies (TWAS) approaches[37,38].

In this study, we conducted a meta-analysis of genome-wide association studies including 12,741 Canadian participants of European-related ancestry from the Canadian Partnership for Tomorrow's Health. We focused our efforts on identifying candidate causal variants and target genes in complex genomic regions known to alter hair colour, by performing a wide range of post-GWAS analyses. Our main outcomes include the identification of multiple candidates causal variants through fine-mapping of significant loci across distinct regions of the genome, including putative causal variants not previously reported for hair colour, and the colocalization of GWAS loci with gene expression and methylation quantitative trait loci (eQTL and meQTL, respectively) using cultured human primary melanocytes. Finally, we conducted transcriptome-wide association studies (TWAS) of hair colour with cultured melanocyte expression data.

## Results

**Hair colour distribution**. 12,996 participants of the Canadian Partnership for Tomorrow's Health (CanPath) project, who were genotyped using different genome-wide genotyping arrays (See Methods for details), self-reported their natural hair colour (before greying) using six possible answers: black ($N = 824$), dark brown ($N = 5,818$), light brown ($N = 4,429$), blonde ($N = 1,410$), red ($N = 306$) hair colour, NA ($N = 30$). After quality control of the genotypes (i.e. exclusion of poor-quality samples and PCA outliers) we kept 12,741 individuals for further analyses.

The distribution of hair colour categories is similar across all provinces sampled (Fig. 1a; Supplementary Data 1), with black and red hair colour being the least frequent categories and brown (light and dark) being the most common one. In addition, on average across all provinces, the proportion of females with black hair colour is lower, whereas other hair colour categories are proportional between sexes (Fig. 1b; Supplementary Data 1), which is similar to what has been previously reported[33,39,40].

**Genome-wide association studies and meta-analyses**. We performed GWAS of hair colour on each of the five genotyping arrays (genotyped and imputed SNPs) using a logistic mixed model on SAIGE (version 0.38)[41]. Specifically, we tested the following models: 1) blonde vs. brown (light and dark) + black hair colour; 2) brown (light and dark) vs. black hair colour; and 3) red vs. brown (light and dark) + black hair colour. For the third model (red vs. brown + black hair colour), the number of individuals with red hair colour on two of the genotyping arrays (GSA 24v1 and Omni 2.5) was less than 20, therefore we excluded these two arrays from the analysis (Supplementary Table 1). In our GWAS models, we included sex, age and the first 10 principal components (PCs) as fixed effects. We did not detect residual population substructure, based on Q-Q plots, in which observed $p$ values did not show an early deviation from the expected p values (Supplementary Figs. 1–3), and the inflation factor (λ: highest value = 1.06; mean = 1.008).

We carried out three meta-analyses using the summary statistics (log of odds ratio and standard error) of each of the GWAS on METASOFT v2.0.1[42]. In total, the number of individuals included on each meta-analysis was: blonde vs. brown + black hair colour: $N = 12,398$; brown vs. black hair colour: $N = 10,990$; and red vs. brown + black hair colour: $N = 10,450$. Q-Q plots of the meta-analyses indicated no inflation of $p$ values (Supplementary Fig. 4). Additionally, the LD Score regression intercepts computed on LDSC (version 1.0.1) were 1.001, 0.994 and 0.999 for

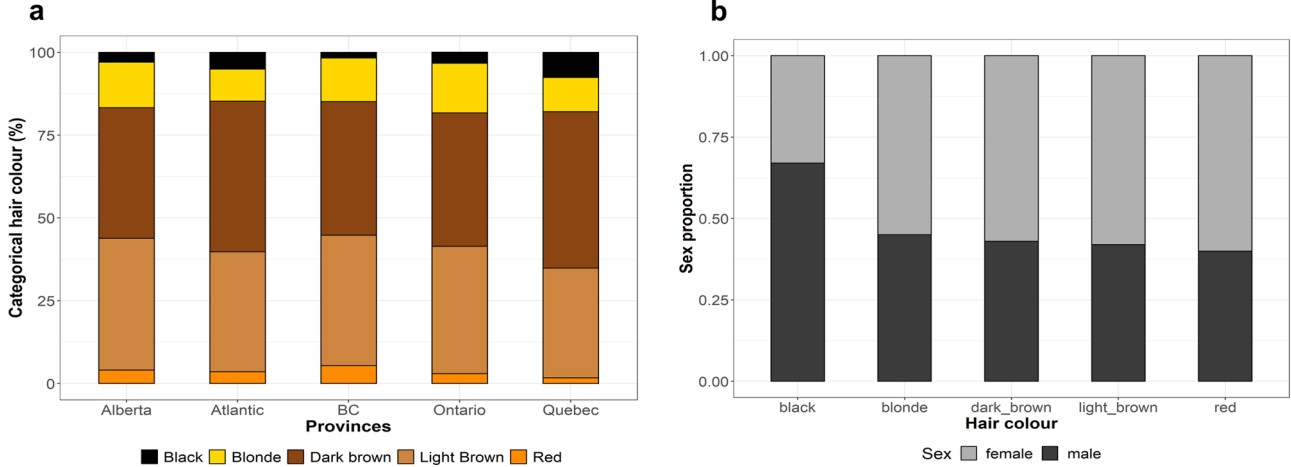

**Fig. 1 Distribution of the hair colour categories in the CanPath cohorts. a** Percentage of each hair colour category, stratified by province. **b** Proportion of sexes across the different hair colour categories.

the three models tested, respectively, indicating no residual confounding bias.

Our meta-analyses identified genome-wide significant loci ($p$ value $\leq$ 1.67e-08) overlapping or near genes known to affect normal pigmentation variation (Fig. 2; Supplementary Figs. 5 and 6). Supplementary Data 2–4 summarise the suggestive and genome-wide associated SNPs for each meta-analysis. On the blonde hair colour meta-analysis (Fig. 2a; Supplementary Fig. 5; Supplementary Data 2), the most significant locus was *OCA2/HERC2* on chromosome (Chr) 15 (lead SNP: rs12913832; $p$ value = 5.03e-141; OR = 0.304; 95% CI = 0.389–0.614). Other genome-wide significant loci overlapped *SLC45A2* (Chr 5), *IRF4* (Chr 6), *TPCN2* (Chr 11), *KITLG* (Chr 12), *SLC24A4* (Chr 14) and *MC1R* (Chr 16). Similarly, on the brown hair colour meta-analysis (Fig. 2b; Supplementary Fig. 5; Supplementary Data 3), the lowest p-value corresponds to a SNP within *HERC2* (rs1129038: $p$ value = 3.36e-52, OR = 0.411; 95% CI = 0.366–0.461), which is in high LD with rs12913832. Other genome-wide significant regions overlap *SLC45A2* (Chr 5) and *IRF4* (Chr 6).

On the red hair colour meta-analysis (Fig. 2c; Supplementary Fig. 5; Supplementary Data 4), the only genome-wide significant locus was *MC1R* (Chr 16, lead SNP: rs12931267, $p$ value = 2.29e-82, OR = 0.014; 95% CI = 0.009–0.022), a gene known for its loss-of-function mutations, which switches the production of eumelanin to pheomelanin[43–45].

Our linear mixed model meta-analysis, in which hair colour categories ranged from blonde up to black (excluding red hair colour), yielded similar results as the logistic mixed models for blonde hair colour (Supplementary Data 5). Particularly, the regions (*TYR, EDNRB, BNC2* and *ASIP*) which were not genome-wide significant in the meta-analyses using logistic mixed models, reached genome-wide significance here (Supplementary Fig. 7). In addition, a locus that did not show up in the logistic mixed model meta-analyses (*ARL15*) was genome-wide significant. This gene has only been previously identified in the UKBB hair pigmentation study[33]. Given that we linearly regressed ordinal categories, we cannot assume that the differences between categories are equal. For this reason, and considering the similar results obtained with the logistic and linear mixed model approaches, we restricted downstream analyses to the meta-analyses based on the binary logistic mixed models.

**Investigating candidate causal variants.** We first investigated if the signals for hair colour across significant loci on our CanPath

meta-analyses were being driven by one or more independent SNPs, by conducting approximate conditional and joint analyses of association on GCTA (GCTA-COJO)[46].

For blonde hair colour, there was one SNP selected for each of the seven genome-wide significant loci (Supplementary Table 2), some of which are known functional SNPs, such as the rs12913832 (Chr 15) and rs12203592 (Chr 6) variants located within enhancers. Similarly, the brown hair colour conditional analysis highlighted one SNP on each of the genome-wide significant loci (Supplementary Table 2). Notably, the SNP selected on chromosome 15 at the *HERC2* locus is rs1129038, which is a SNP in high LD with rs12913832.

We identified five independent SNPs for red hair colour on chromosome 16, overlapping or near the *MC1R* region. A few of these markers show evidence of heterogeneity (Supplementary Table 2), but have a similar effect size and p-value on the fixed and random-effects models. This result is in agreement with previous studies, as it is well known that there are multiple loss-of-function mutations on this region affecting hair colour variation[33,44,47,48]. Finally, we performed the approximate conditional analyses a second time using a different reference LD matrix from a subset of our CanPath samples (See Methods for details). Notably, there were no overall differences in the results obtained with either matrices for blonde or brown hair colour, in respect to the independent SNPs per locus and overall summary statistics (Supplementary Data 6). In the case of red hair colour, there were a total of seven SNPs independently associated in the locus on chromosome 16.

We then carried out Bayesian fine-mapping of these loci to identify candidate causal SNPs using FINEMAP v1.4[49]. Compared to GCTA's stepwise conditioning approach, which depends on arbitrary p-value thresholds, Bayesian fine-mapping quantifies the probability of causality by jointly modelling simultaneous effects of multiple SNPs, and considers genotype probabilities for calculating LD[36]. Each credible set contains a minimum set of candidate causal SNPs with a probability of at least 95%, and we kept the candidate causal variants from each credible set that had considerable evidence of causality (i.e. SNPs with logBF $\geq$ 2), and included their respective annotations with SNPNexus[50,51] (See Methods for details). Based on the combined evidence of fine-mapping and annotation, we defined the candidate causal variants with strong evidence of causality as the most likely candidate causal variants. We have summarised the results for each hair colour model (Supplementary Table 3; Supplementary Data 7–9).

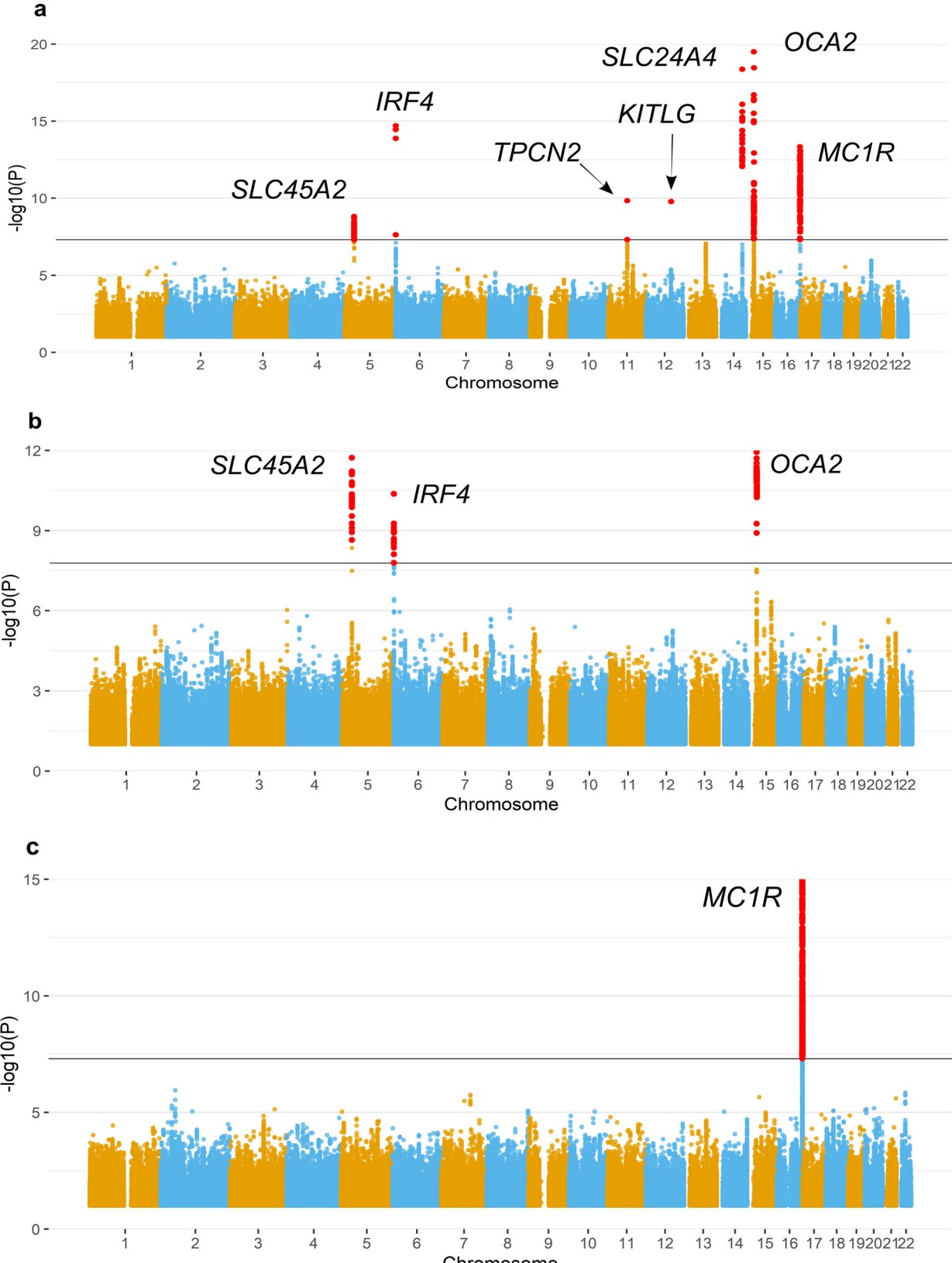

**Fig. 2 Manhattan plots of hair colour meta-analyses based on logistic mixed models. a** Blonde vs. brown + black hair colour ($n = 12{,}398$ individuals). **b** Brown vs. black hair colour ($n = 10{,}990$ individuals). **c** Red vs. brown + black hair colour ($n = 10{,}450$ individuals). The continuous black line denotes the genome-wide significant threshold ($p = 1.66e\text{-}8$). Markers in red are genome-wide significant. The Y-axis has been limited to truncate strong signals of association. The full figure is available as Supplementary Fig. 8.

Bayesian fine-mapping analyses highlighted known functional pigmentation SNPs, in line with the results obtained with GCTA-COJO, such as rs12203592 within *IRF4* (Chr 6), which is the most likely candidate causal SNP in the locus, with high posterior probability (identified in the blonde and brown hair colour analyses) (Supplementary Fig. 8; Supplementary Table 3). On the *SLC45A2* locus (Chr 5), the known missense SNP rs16891982 is present within the 95% credible set, although the SNP with the highest posterior probability is rs35391. On the *SLC24A4* locus (Chr 14), the marker highlighted by GCTA-COJO (rs12896471) is among the most likely candidate causal SNPs. This marker is in perfect LD ($r^2 = 1$) with a SNP previously associated with blonde hair colour (rs12896399)[52], and both appear in the same 95% credible set.

Different from GCTA-COJO, on the *TPCN2* region (Chr 9), we identified the missense SNP rs3829241 as one of the most likely candidates of causality in the locus (Supplementary Fig. 9). Additionally, the SNP rs72932523, highlighted by GCTA-COJO, also has considerable evidence of causality in an independent credible set (Supplementary Fig. 9). This marker is in high LD ($r^2 = 0.83$) with the missense SNP rs72928978. Another two known missense variants (rs3750965 and rs35264875) within *TPCN2* appear in the same 95% credible set as rs3829241, but with a $\log_{10}BF < 2$, suggesting little evidence of causality.

On the *HERC2/OCA2* region (Chr 15), the only SNP with considerable evidence of causality for blonde hair colour, based on its posterior probability and annotation, is the known regulatory SNP rs12913832 (PIP = 0.978). In contrast, there are two causal signals identified for brown hair colour highlighted by FINEMAP (Supplementary Fig. 10). Similar to the blonde hair colour analysis, one of the credible sets include rs12913832 (PIP = 0.577). The second credible set includes candidate causal SNPs within *OCA2* with low PIP (<0.5), in which one of them has $\log_{10}BF > 2$ (rs7168800) (Supplementary Table 3).

Finally, the *MC1R* region (Chr 16) was associated with both blonde and red hair colour. FINEMAP highlighted three causal signals in the locus for blonde hair colour and ten causal signals for red hair colour. The credible sets include known missense SNPs within or near *MC1R* (rs1805005, rs1805007, rs1805008, rs1805009), in which rs1805005 had the highest PIP and $\log_{10}BF$ on the blonde hair colour analysis (PIP = 0.984; $\log_{10}BF$ = 4.970) (Supplementary Fig. 11); this same SNP does not appear in the red hair colour credible sets. Similarly, the missense SNP rs1805009 is not present in the blonde hair colour credible sets. Additionally, most candidate causal SNPs in the credible sets for red hair colour had a PIP > 0.9 and $\log_{10}BF$ well above 2 (Supplementary Data 9). When comparing the *MC1R* locus results obtained with GCTA with these results, we noticed an overlap of the candidate causal SNPs, specifically of the missense SNPs, whereas the synonymous SNPs highlighted by each method differ.

**Exploring gene expression and methylation using cultured melanocyte data**. We conducted colocalization analyses of the GWAS meta-analyses genome-wide signals using hyprcoloc[53] with gene expression and methylation *cis*-QTLs (eQTLs, meQTLs, respectively) to explore the putative regulatory role of the SNPs identified in our hair colour GWAS and identify candidate genes (Table 1—See Methods for details). We observed colocalization of GWAS signals with eQTLs of *SLC45A2* and *OCA2* (marked by SNPs rs35391 and rs12913832, respectively). Additionally, we also observed colocalization of a GWAS signal with an eQTL of *SLC24A4*, with a regional probability = 1, but there was no candidate SNP selected, suggesting limited evidence of a single colocalized SNP between the traits. In addition, we

identified five GWAS loci colocalized with meQTLs, associated with methylation of CpGs near *MC1R*, *OCA2*, *IRF4*, *SLC45A2* and *SLC24A4*. Notably, we found GWAS colocalization with both eQTL and meQTL in three of these loci (*SLC24A4*, *SLC45A2* and *OCA2* regions). In the case of *SLC24A4*, the SNP rs8022442 is a meQTL for CpG probes, cg11086312 and cg10004481. On the *OCA2* locus, rs12913832 is a meQTL for CpG probes located upstream and downstream of the SNP within *HERC2* (cg05271345, cg25622125 and cg27374167). Furthermore, the SNP rs35391 is a meQTL for the CpG probes in the first exon of *SLC45A2* region (cg14189614 and cg04302388).

We did not find colocalization of QTLs on or near the gene *MC1R* for red hair colour. Given the current evidence, this is likely explained by the fact that known loss-of-function polymorphisms within *MC1R* lead to red hair colour, therefore, they have a direct functional role on the translated protein. However, there is a possibility that we might be missing eQTLs beyond the 500 kb tested region. Nonetheless, we did identify colocalizing meQTLs (rs258322) for blonde hair colour, associated with the CpG methylation near *MC1R*: on or near *CDK10*, *GAS8* and *DPEP1* (cg05714116, cg06907930 and cg00996377), which may point at an independent regulatory region associated with blonde hair colour, apart from the known missense SNP within *MC1R* (rs1805005)[39]. Additionally, we observed a colocalized signal between GWAS and meQTL near *IRF4* for both blonde and brown hair colour (cg23785612). Finally, neither of these loci (i.e., *MC1R*, *IRF4*) harbour corresponding eQTLs.

Lastly, we conducted transcriptome-wide association studies (TWAS) using the GWAS summary statistics, and we imputed the expression profile based on the CanPath cohort LD, from the expression weights calculated from the cultured melanocytes RNA-seq data[54]. Similar to the colocalization results, the decreased expression of *OCA2* and *SLC24A4* was significantly associated with blonde hair colour (Table 2; Supplementary Fig. 12). In contrast, the increased expression of *EDNRB* was associated with blonde hair colour, which was not identified through colocalization analyses. In this case, the direction of effect in both the eQTL and GWAS is negative, which is the opposite of what was expected, given that the protein encoded by *EDNRB* is involved in melanocyte development and it induces melanocyte proliferation[55]. This discrepancy could be due to different direction of effects in skin and hair melanocytes, similar to the inverse effect of *IRF4* effect on hair and skin pigmentation[56]. However, further investigation of the *EDNRB* expression patterns in the hair bulb is needed to provide a clear explanation.

The decreased expression of the gene *RIN3* near *SLC24A4* was also significantly associated with blonde hair colour. After performing conditional TWAS analysis to check if these two signals were independent from each other, we found that the gene *RIN3* signal is not independent from that of the nearby gene *SLC24A4* (Supplementary Fig. 13). Lastly, the decreased expression of *CDK10* was significantly associated with red hair colour, which contrasts our colocalization results, in which we did not identify eQTLs for red hair colour. The gene *CDK10* is near *MC1R*, but to the extent of our knowledge, this gene is not implicated in pigmentation. However, it is important to note that TWAS results do not imply gene causality, and further evidence and replication of this signal is needed to have a conclusive result.

**Biological pathway gene set enrichment analysis**. We performed a gene set analysis to identify relevant biological pathways, using the results of our candidate causal SNP analysis using FINEMAP (Supplementary Table 4; Fig. 3; Supplementary Data 10), for all hair colour loci jointly. Focusing on the gene ontology (GO)

**Table 1 Colocalization results of expression and methylation *cis*-QTLs from cultured melanocytes (eQTL and meQTL, respectively) with GWAS SNPs on each hair colour category.**

| Chromosome | Candidate SNP | Posterior probability | Regional probability | Posterior explained by SNP | Gene/methylation annotation | QTL |
|---|---|---|---|---|---|---|
| Blonde Hair Colour | | | | | | |
| 5 | rs35391 | 0.83 | 0.85 | 0.43 | *SLC45A2* | eQTL |
| 5 | rs35391 | 0.98 | 0.99 | 0.54 | *SLC45A2*\|OpenSea | meQTL |
| 5 | rs35391 | 0.96 | 0.97 | 0.55 | OpenSea | meQTL |
| 6 | NA | NA | 0.95 | NA | OpenSea | meQTL |
| 14 | NA | NA | 1.00 | NA | *SLC24A4* | eQTL |
| 14 | rs8022442 | 0.99 | 1.00 | 1.00 | OpenSea | meQTL |
| 14 | rs8022442 | 0.95 | 0.95 | 1.00 | *SLC24A4* (Body) \| OpenSea | meQTL |
| 15 | rs12913832 | 1.00 | 1.00 | 1.00 | *OCA2* | eQTL |
| 15 | rs12913832 | 0.99 | 0.99 | 0.94 | AC090696.2 | eQTL |
| 15 | rs12913832 | 1.00 | 1.00 | 0.99 | *HERC2* (Body) \| OpenSea | meQTL |
| 15 | rs12913832 | 0.98 | 0.99 | 0.98 | *HERC2* (Body) \| S_Shelf | meQTL |
| 15 | rs12913832 | 0.98 | 0.98 | 0.98 | *HERC2* (Body) \| S_Shore | meQTL |
| 16 | rs258322 | 1.00 | 1.00 | 1.00 | *CDK10* (TSS1500)\|N_Shore | meQTL |
| 16 | rs258322 | 1.00 | 1.00 | 1.00 | LOC100130015 (Body); *GAS8* (3′ UTR)\|OpenSea | meQTL |
| 16 | rs258322 | 0.94 | 0.94 | 1.00 | *DPEP1* (5′ UTR)\|OpenSea | meQTL |
| Brown Hair Colour | | | | | | |
| 5 | rs35391 | 0.83 | 0.85 | 0.57 | *SLC45A2* | eQTL |
| 5 | rs35391 | 0.98 | 0.99 | 0.65 | *SLC45A2* (1st Exon) \| OpenSea | meQTL |
| 5 | rs35391 | 0.97 | 0.97 | 0.66 | Open Sea | meQTL |
| 6 | rs7773997 | 0.94 | 1.00 | 0.94 | Open Sea | meQTL |
| 15 | rs12913832 | 1.00 | 1.00 | 0.98 | *OCA2* | eQTL |
| 15 | rs1129038 | 0.99 | 0.99 | 0.54 | AC090696.2 | eQTL |
| 15 | rs12913832 | 1.00 | 1.00 | 0.87 | *HERC2* (Body) \| OpenSea | meQTL |
| 15 | rs12913832 | 0.98 | 0.98 | 0.73 | *HERC2* (Body) \| S_Shelf | meQTL |
| 15 | rs12913832 | 0.97 | 0.97 | 0.74 | *HERC2* (Body) \| S_Shore | meQTL |

We show colocalized SNPs with a posterior probability of ≥0.8. We tested all the significant eQTL genes or meQTL probes within ±250 kb regions flanking the GWAS lead SNPs. The Gene/Methylation Annotation indicates the location of CpG probes with respect to the nearest gene, as well as relative to CpG island. NA = limited evidence of a single SNP driving the colocalization.

**Table 2 Genome-wide significant genes in the TWAS of the hair colour categories: blonde, brown and red.**

| Gene | Chr | GWAS best SNP | GWAS Z-score | eQTL best SNP | eQTL Z-score | # of SNPs | # weighted SNPs | TWAS Z-score | TWAS p-value |
|---|---|---|---|---|---|---|---|---|---|
| Blonde Hair Colour | | | | | | | | | |
| *EDNRB* | 13 | rs7330412 | −5.4 | rs7330412 | −3.89 | 1204 | 1 | 5.404 | 6.52E-08 |
| *RIN3** | 14 | rs12896399 | 11.51 | rs12893289 | −6.14 | 986 | 4 | −11.3069 | 1.21E-29 |
| *SLC24A4** | 14 | rs12896399 | 11.51 | rs61977801 | −6.68 | 933 | 11 | −11.1083 | 1.14E-28 |
| *OCA2* | 15 | rs12913832 | −25.28 | rs12913832 | 6.24 | 489 | 1 | −25.282 | 5.04E-141 |
| Brown Hair Colour | | | | | | | | | |
| *OCA2* | 15 | rs1129038 | −15.2 | rs12913832 | 6.24 | 489 | 1 | −15.203 | 3.38E-52 |
| Red Hair Colour | | | | | | | | | |
| *CDK10* | 16 | rs1805007 | 18.67 | rs11538871 | −5.69 | 987 | 5 | −7.23615 | 4.62E-13 |

Genome-wide significant threshold: p-value ≤ 4.17e-06. GWAS/eQTL best SNP is the most significant SNP on each analysis. *Chr* chromosome. *Signals are not independent from each other, as evidenced by conditional TWAS.

processes, most gene sets correspond to pigmentation processes (e.g., developmental pigmentation, melanin biosynthetic process, melanocyte differentiation) or are a parental process of a pigmentation-related process (e.g., secondary metabolite biosynthetic process, phenol-containing compound biosynthetic process).

Notably, the DNA repair process includes genes surrounding the *MC1R* gene (i.e., *FANCA*, *SPIRE2*), as well as *HERC2*. It is well known that *MC1R* signalling reduces UV-induced DNA damage by mediating a cascade of reactions after the activation of cAMP-dependent Protein Kinase A (PKA)[57]. Likewise, loss-of-function mutations on *MC1R* diminish the UV-induced DNA damage repair[57]. *HERC2* is known to be involved in DNA repair induced by UV[58], but the expression of this gene likely does not

play a role in hair colour variation, as the associated SNPs are within introns, serving as enhancers to regulate the expression of *OCA2*.

Pigmentation traits, such as red/blonde hair, fair skin and response to sun exposure, are risk factors for skin cancer (melanoma and nonmelanoma)[27,59]. We assessed if our genome-wide significant signals associated with hair colour have also been significantly associated with sun tanning (i.e., response to UV radiation exposure), melanoma, basal cell carcinoma (BCC) and squamous cell carcinoma (SCC), based on the databases available in the NHGRI-EBI GWAS Catalog[60]. We tested a total of 327 SNPs across all phenotypes (melanoma: 153, BCC: 103, SCC: 29, and sun tanning: 42 SNPs), of which 21 unique SNPs overlap our GWAS hits (melanoma: 13, BCC: 5, SCC: 9, and sun tanning: 5 SNPs).

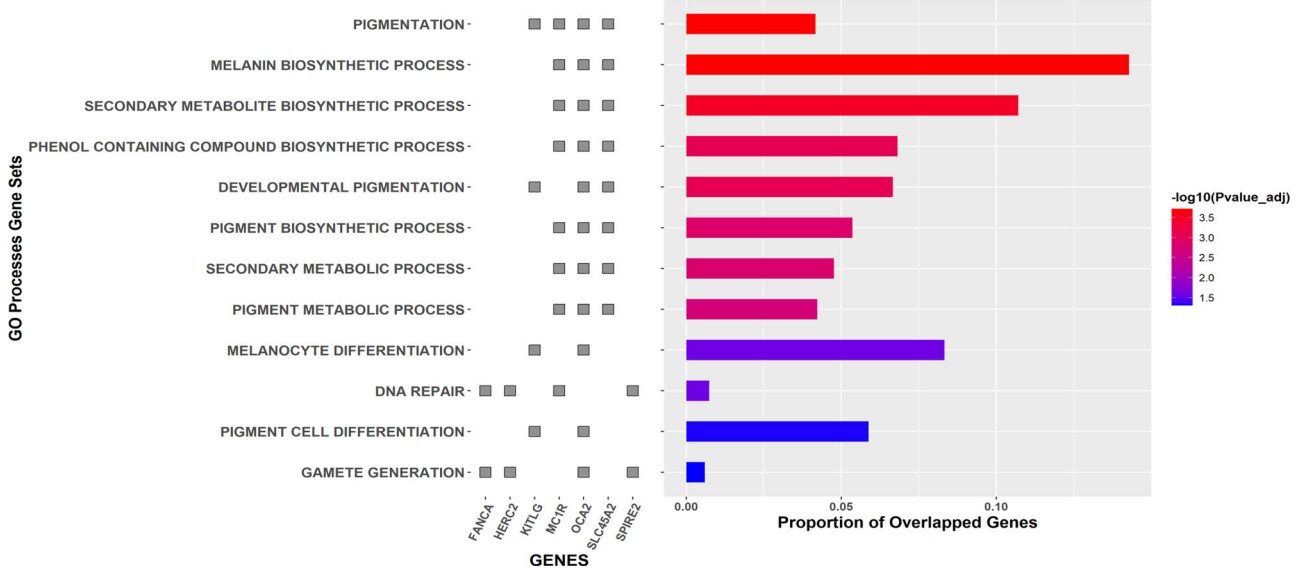

**Fig. 3 Enrichment pathway analysis of genome-wide significant hair colour signals.** Analysis performed with FUMA using Gene Ontology processes for all genes significantly associated with hair colour, as indicated by FINEMAP results ($n = 13$ genes).

These SNPs are distributed across key pigmentation loci that overlap genes, such as *SLC45A2, IRF4, SLC24A4, OCA2, HERC2* and *MC1R* (Supplementary Table 5). However, we note that an overlap of significant signals is not indicative of a shared causal signal between traits, therefore the biological relevance of these loci on skin cancer should be functionally investigated.

## Discussion

In this study, we conducted meta-analyses of genome-wide association studies of hair colour on a Canadian cohort of European ancestry. Furthermore, we performed several post-GWAS analyses to investigate in more detail the genetic architecture of this trait, by focusing on the genome-wide associated loci. By applying statistical fine-mapping methods, we identified pigmentation loci harbouring more than one causal signal with high confidence, similar to what has been previously observed[5,19,61]. Additionally, we have incorporated colocalization analyses of expression and methylation QTLs, as well as TWAS, both of which provide insights on the regulatory mechanisms of hair colour variation and pigmentation biology.

We note that hair colour is a naturally quantitative trait, which shows substantial variation as a result of differences in the amount of melanin in the hair, as well as differences in the ratio between the two types of melanin, eumelanin and pheomelanin. In an ideal scenario, quantification of the phenotype on a large sample would yield more accurate association results given the nature of the trait. However, recent studies from large-scale biobanks have shown a drastic increase in power to detect significant associations when the sample sizes increase, even if a self-reported, qualitative approximation of hair colour is used[5,32,33].

To investigate the genetic architecture of hair colour categories, we relied on our binary logistic mixed model analyses, in which we identified several known genes involved in pigmentation variation, such as *SLC45A2, KITLG, SLC24A4, OCA2, MC1R*, among others. However, by using a linear GWAS model based on four eumelanin ordinal categories (i.e. blonde, light brown, dark brown, black), we replicated the signal of the gene *ARL15* that was previously identified in a recent UKBB study[33] (Supplementary Data 5).

We identified through fine-mapping analysis known pigmentation variants, which are likely candidates of causality, such as

the missense SNP rs16891982 on *SLC45A2*, rs12821256 near *KITLG*, rs12203592 on *IRF4* and rs12913832 on *HERC2*, the latter three located within known enhancers affecting the transcription of *IRF4, KITLG,* and *OCA2*, respectively[56,62–65].

Our fine-mapping analyses highlighted several regions that harbour more than one causal signal. On the *TPCN2* locus, we identified two likely causal signals that include or are in high LD with missense SNPs: rs3829241 and rs72932523 (in LD with rs72928978). It is known that *TPCN2* encodes a cation transporter channel in melanosomes that regulates the pH of melanosomes to downregulate melanogenesis[66,67]. There are three nonsynonymous SNPs (rs3829241, rs35264875, and rs3750965) associated with blonde hair colour reported in the GWAS literature[33,52]. Two of these nonsynonymous SNPs (rs3829241 and rs35264875) have been experimentally shown to modify the pH of the melanosome, an important factor for tyrosinase's activity[68].

Additionally, it is possible that the missense SNP rs72928978, which has been recently associated with eye colour[69], may also have a functional effect on the TPC2 protein, as predicted by SIFT and Polyphen (deleterious and possibly damaging, respectively). In addition, the A-allele of rs72928978 shows high population differentiation: it is common only in European and European admixed populations of the 1KGP (mean MAF in EUR = 0.14) and it is absent from other continental populations of the 1KGP (Supplementary Fig. 14). Therefore, future experimental analyses on the missense SNP rs72928978 may provide more details regarding its functional effect on melanin synthesis.

The *HERC2* rs12913832 SNP is one of the most important determinants of eye colour variation in human populations, and more particularly blue vs. non-blue eye colour. This SNP is located within an enhancer on an intron of *HERC2*, which regulates the expression of the downstream gene *OCA2*[62]. We report here that rs12913832 is both an expression and methylation QTL in melanocytes. However, we cannot be certain of a correlation between the meQTL target CpG and *OCA2* expression, given the current evidence.

We were particularly interested in studying in more detail the *OCA2/HERC2* region due to previous evidence pointing to several independent candidate variants in this region[5,18,19]. In addition to the signal described above for rs12913832, our fine-mapping

results indicate that there is evidence of another candidate causal signal associated with brown hair colour on the *OCA2* gene. The second credible set comprises SNPs within *OCA2*, including the intronic SNPs rs72714118 and rs72714121 (PIP = 0.12 and 0.10, respectively), both of which overlap histone marks identified in foreskin melanocytes (H3K27ac, H2K4me1) associated with enhancer signals[70]. This result contrasts to observations by Adhikari *et al.*, in which they detect a secondary variant associated with eye and skin pigmentation on the *HERC2/OCA2* locus, but not with hair colour[5].

The red hair colour phenotype is mainly a consequence of missense polymorphisms on the *MC1R* locus, some of which have high penetrance and are termed "*R*" variants (rs1805007, rs1805008, rs1805009, rs1805006) and others that have low penetrance and are termed "*r*" variants (rs1805005, rs2228479, rs885479). Compared to "*R*" variants, which impair the function of the protein and lead to the synthesis of pheomelanin in a homozygote or compound heterozygote state, "*r*" variants only reduce the protein's efficiency, leading to low levels of eumelanin synthesis[17,71]. We have identified here candidate causal R and r variants associated with red hair colour (rs1805008, rs1805009), but also with blonde hair colour (rs1805005, rs1805007, rs1805008). The SNP rs1805005 is one of the most likely candidate causal SNPs, although there is a high probability of additional causal signals in the region. In fact, a SNP within *CDK10* (rs258322) is a colocalizing meQTL for blonde hair colour. The meQTL is in moderate LD with a candidate causal SNP with $\log_{10}BF > 2$ (rs75570604; $r^2 = 0.62$). These results provide insights about possible regulatory variants leading to blonde hair colour.

Within the solute carrier family, there are at least three transmembrane proteins involved in ion transport (*SLC24A5*, *SLC45A2* and *SLC24A4*), which are also involved in normal pigmentation variation. Light skin pigmentation in people of European ancestry is driven mainly by two nonsynonymous mutations (rs1426654 and rs16891982) on *SLC24A5* and *SLC45A2*, respectively[2,14,64,72]. Notably, our colocalization analyses suggest that there may be additional SNPs in the *SLC45A2* region regulating the expression of the gene.

Several reports have identified the association of *SLC24A4* with hair and eye colour in the same population[2,47,73–75], highlighting an upstream SNP (~15 kb from the transcription start site) with the largest effect (rs12896399), which is a common variant in all 1KGP continental populations, except in African populations. Based on the Genotype-Tissue Expression (GTEx) Project, rs12896399 is significantly associated with the expression of *SLC24A4* in skin tissue (skin not sun-exposed; *p* value = 2.3e-6). However, to the extent of our knowledge, there is no clear evidence of the molecular process by which this SNP regulates the expression of *SLC24A4*. By conducting colocalization and TWAS analyses, we identified both an eQTL and meQTL in the *SLC24A4* locus, in which the candidate meQTL is in moderate LD with the fine-mapped candidate causal SNP rs12896471 ($r^2 = 0.65$).

Our colocalization results highlighted meQTLs for blonde hair colour, associated with the methylation of CpGs near known pigmentation genes (i.e., *MC1R*, *IRF4*). These loci do not harbour colocalizing eQTLs, which suggests that other mechanisms may be involved, such as *trans*-QTLs[76], which were not considered in the current analysis. Alternatively, it is possible that some CpG probes capture the status of poised enhancers (i.e., enhancers in a *latent* state), which may not yet have any influence on gene expression in actively growing melanocytes. This is a possible scenario, given that the melanocytes used in the QTL analyses were from newborns. However, none of the candidate colocalized SNPs in these loci (rs258322 and rs7773997) are eQTLs of *MC1R* and *IRF4*, respectively, in adult skin tissue based on the GTEx Project. Experimental histone modification marker assays may

provide support for the alternative hypothesis, as it is known that poised enhancers lose H3K27me3 and acquire acetylation at the same amino acid residue upon activation[77].

Several of the genome-wide significant SNPs we identified in the CanPath cohort overlap SNPs associated with different types of skin cancer and response to UV radiation, and it is particularly the case for melanoma. Additionally, a few of the overlapped SNPs are also colocalized eQTLs and/or meQTLs (e.g., rs4904871), which highlights the importance of investigating the genetic and epigenetic mechanisms involved in the pigmentation pathway, such as hair colour. In fact, a recent study has provided a first glance into the epigenetic mechanisms (i.e., DNA methylation and gene expression) of pigmentation genes mediating skin cancer using whole blood tissue[78]. By applying summary-based Mendelian randomisation and colocalization analyses, they colocalized 9 DNA methylation sites (DNAm) with pigmentation traits (skin cancer, hair colour and sun exposure), as well as with the expression of genes.

We followed-up their QTLs (Table 3 of Bonilla et al.[78]) on our colocalization results, but none of their SNPs was present in our colocalization results. The differences may lie in the fact that we used cultured melanocytes, which provide a cell-specific expression and methylation profile, best suited for the traits being tested[76]. However, it is relevant to note that the authors also successfully colocalized a DNAm site near the gene *CDK10* with blonde hair colour, as well as with several skin cancer traits (i.e., melanoma, basal cell carcinoma), which further reinforces the evidence we reported here, regarding putative regulatory variants in that locus.

Overall, our results indicate that the performance of GCTA-COJO and FINEMAP is largely concordant, although the credible sets and posterior probabilities computed by FINEMAP, combined with appropriate annotations, provide a broader approach to prioritise the most likely causal variants for further functional validation. It is worth noting that our analyses focused on SNPs due to the nature of the data and imputation approach, therefore we may be missing important structural variants that contribute to pigmentation variation, such as small indels. Finally, the lack of sex chromosome data also poses a limitation in the current study.

In conclusion, by taking advantage of a relatively large cohort like the CanPath, we conducted GWAS meta-analyses of hair colour in which we identified candidate causal variants and provided insights into the genetic architecture that modulates hair colour variation. Many of these variants also affect other pigmentation traits, such as normal skin pigmentation variation, tanning response, as well as different types of skin cancer. We took advantage of expression and methylation data to characterise nonprotein-coding GWAS hits, and we believe that further experimental assays that include other epigenetic elements will provide further details on the genomic mechanisms regulating pigmentation variation. Our results provide insights on the general mechanisms regulating pigmentation biology in humans.

## Methods

**Canadian partnership for tomorrow's health participants**. This study was approved by the University of Toronto Ethics Committee (Human Research Protocol # 36429) and data access was granted by the Canadian Partnership for Tomorrow's Health (Application number DAO-034431). All relevant ethical regulations were followed, and informed consent was obtained from CanPath participants. The samples in this study correspond to a subset of 12,996 individuals from the Canadian Partnership for Tomorrow's Health (CanPath), which were sampled in different provinces: Alberta ($N = 969$; 7.45%), Atlantic Coast Provinces (i.e., New Brunswick, Newfoundland, Nova Scotia and Prince Edward Island) ($N = 937$; 7.21%), British Columbia ($N = 986$; 7.59%), Ontario ($N = 941$; 7.24%), and Quebec ($N = 9,163$; 70.51%). We selected individuals who self-reported having European-related ancestry and for whom self-reported hair colour was available. Among all participants included here, 53.78% were females and the average age was 53 years old (SE ± 7.85).

**Genotyping of participants and quality control**. An overview of the methodological workflow is shown in Supplementary Fig. 15. Individuals who self-reported as having European-related ancestry were genotyped between 2012 and 2018 using five different genotyping array chips: (i) Axiom 2.0 UK Biobank (Affymetrix) ($N = 4,821$), (ii) Global Screening Array (GSA, Illumina) 24v1 ($N = 438$), (iii) 24v2 + MDP ($N = 2,594$), (iv) 24v1 + MDP ($N = 4,617$), and (v) Omni 2.5 (Illumina) ($N = 526$) by the Canadian Partnership for Tomorrow's Health (CanPath) project. The number of single-nucleotide polymorphisms (SNPs) of these chip arrays ranges between 626,377 and 2,349,746 SNPs.

We performed genotype quality control for each array chip separately by first filtering out variants that deviated in minor allele frequency >0.2 from the 1000 Genomes Project Phase 3 European sample (1KGP-EUR), GC/TA variants with minor allele frequency >0.4 in the 1KGP-EUR and flipping alleles according to the 1KGP-EUR, using a Perl script (version 4.2)[79]. Afterwards, we used PLINK (version 1.9)[80,81] to filter out variants with minor allele frequency <1%, high missing genotyping rate (–geno 0.05), high missing individual rate (–mind 0.05) or variants that significantly deviated from the Hardy-Weinberg Equilibrium (HWE) (–hwe 1e-06). Then, we also identified second-degree relatives (–genome, PI_HAT > 0.2) using a pruned set of variants in linkage disequilibrium (LD) (–indep-pairwise 100 10 0.1), and filtered out, from each pair, the individual with the lowest genotyping rate. We performed a Principal Components Analysis (PCA) of a pruned set of common variants of our study samples projected on the 1KGP European Phase 3 samples on PLINK (version 1.9)[80,81] (Supplementary Figs. 16 and 17). Finally, we performed a PCA with the full 1KGP Phase 3 samples and removed individual outliers that did not cluster within the European sample of the 1KGP by inspecting the first three principal components (total PCA outliers across genotyping arrays = 81). Amongst the outliers, 63 individuals are from Quebec, 8 from British Columbia, 5 from the Atlantic Provinces, 5 from Alberta and none from Ontario.

The final number of SNPs (n) and individuals (N) remaining after quality control for each chip array were: (i) Axiom 2.0 UK Biobank (n = 630,508; $N = 4,745$), (ii) GSA 24v1 (n = 558,183; $N = 438$), (iii) GSA 24v2 + MDP ($n = 596,061$; $N = 2,553$), (iv) GSA 24v1 + MDP ($n = 596,061$; $N = 4,480$), and (v) Omni 2.5 ($n = 2,081,743$; $N = 525$). (Supplementary Table 6), yielding a total number of 12,741 individuals from different provinces in Canada.

**Imputation of Genotypes**. Each genotyping array was first phased with EAGLE2 (version 2.0.5)[82] using the Sanger Imputation Server[83]. After phasing, samples on each genotyping array were imputed on the Sanger Imputation Server using the positional Burrows-Wheeler transform (PBWT) algorithm[84] and the Haplotype Reference Consortium (HRC) release 1.1 dataset as reference[83]. The HRC includes ~64,000 haplotypes and ~40,000,000 autosomal SNPs of ~32,000 individuals predominantly of European ancestry, which makes it ideal for the imputation of our datasets, which are of European-related ancestry. Post-imputation quality control consisted of filtering out variants with INFO score <0.3. Briefly, The INFO score is a measure of the imputation certainty across samples, in which INFO = 1 indicates complete certainty. We also filtered out variants with MAF <0.01, missing genotyping rate >5% or variants that significantly deviated from the HWE. The final number of markers included in the GWAS for each genotyping array were: (i) Axiom 2.0 UK Biobank = 6,880,138, (ii) GSA 24v1 = 6,185,935, (iii) GSA 24v2 + MDP = 6,214,597, (iv) GSA 24v1 + MDP = 6,204,261, and (v) Omni 2.5 = 7,391,256.

**Phenotyping**. Participants of the CanPath answered a questionnaire that included self-report on natural hair colour (before greying) using the following discrete categories: black, dark brown, light brown, blonde, red hair colour or NA. These categories were then transformed into binary categories using R (version 3.5.1)[85] to build logistic mixed models to compare the presence (1) or absence (0) of: 1) blonde vs. brown (light and dark) + black hair colour; 2) brown (light and dark) vs. black hair colour; and 3) red vs. brown (light and dark) + black hair colour, similar to the approach used by Morgan and colleagues[33]. Supplementary Table 1 shows the number of individuals on each hair colour category by genotyping array, and Supplementary Fig. 15 shows an overview of the methodology. In addition, participants also reported their age and sex.

**Genome-wide association studies (GWAS) and meta-analyses**. Genome-wide association studies of hair colour were performed for each genotyping array with binary logistic mixed models on SAIGE (version 0.38)[41], using an additive genetic model (i.e. the effect size is a linear function of the number of effect alleles), and considering the genotypes' dosages. Specifically, the three hair colour models used were: 1) blonde vs. brown (light and dark) + black hair colour; 2) brown (light and dark) vs. black hair colour; and 3) red vs. brown (light and dark) + black hair colour. We performed a PCA of a pruned set of genotyped variants for each genotyping array after quality control, keeping only SNPs with MAF > 0.05 and excluding regions of high LD, using PLINK (version 1.9)[80,81]. We included 10 PCs as fixed effects in the logistic mixed models for all genotyping arrays. We also included in the model sex and age as fixed effects. Additionally, we included as random effects a genetic relationship matrix of independent markers to account for subtle structure, computed on PLINK (version 1.9)[80,81]. We did not perform a

GWAS for the presence/absence of red hair colour in the two small samples (Omni 2.5 and GSA 24v1) due to the low number of cases ($N < 20$). To evaluate the case of residual population substructure, we computed the inflation factor (λ) and visualised the expected vs. observed p values using Q-Q plots on R (version 3.5.1)[85].

We performed a meta-analysis for each hair colour model (blonde, brown and red hair colour) using the beta coefficient (i.e. log(odds ratio)) and standard error (SE) of each study on the software METASOFT (version 2.0.1)[42]. METASOFT conducts a meta-analysis using a fixed-effects model (FE), which works well when there is no evidence of heterogeneity (i.e. assumes the same effect size across studies), and an optimised random-effects model (RE2), which works well when there is evidence of heterogeneity among studies[42]. Additionally, METASOFT computes two estimates of statistical heterogeneity, Cochran's Q statistic and $I^2$ statistic[86]. We considered a SNP as heterogeneous across studies at an alpha level of 0.05 and $K-1$ degrees of freedom, where $K$ is the number of studies included in the meta-analysis. Similarly, values of $I^2 > 50\%$ are considered to represent notable heterogeneity[87].

After conducting the meta-analyses results, we generated Manhattan and Q-Q plots using the qqman[88] and ggplot2[89] R packages. We used a genome-wide significant threshold of 1.67e-08 (i.e., 5e-08/3) to account for the three models tested. We focused our results on the fixed-effects model, but we also report the RE2 on the summary statistics of the top signals as Supplementary Data 2–4, and compared the statistical significance between the models when there was evidence of heterogeneity based on Cochran's Q and $I^2$ statistics. We performed LD Score regression on LDSC[90] (version 1.0.1) on each of the meta-analyses summary statistics to evaluate possible inflation, using the LD Scores from the European population of the 1000 Genomes Project.

Additionally, we conducted GWAS and meta-analyses of hair colour using a linear mixed model approach, similar to the methods used in previous hair pigmentation studies[5,32]. We coded the hair colour categories as a quantitative trait, spanning from low to high eumelanin (1 = blonde, 2 = light brown, 3 = dark brown, 4 = black). For this analysis, we excluded red hair colour, given that red hair colour is characterised by its high levels of pheomelanin, and does not fall within the low to high eumelanin spectrum[91]. We performed the GWAS on GCTA 1.26.0[92,93], including as fixed-effects sex, age and the first 10 PCs and a genetic relationship matrix as random effects. Subsequently, we performed a meta-analysis on METASOFT (version 2.0.1)[42]. We generated Manhattan plots for the meta-analysis results.

**Annotation of significant loci**. We used the web-based programme SNPnexus[50,51] to annotate the genome-wide significant signals ($p$-value ≤ 1.67e-08) from each meta-analysis. Specifically, gene and variant type annotation were done using the University of California Santa Cruz (UCSC) and Ensembl databases (human genome version hg19); assessment of the predictive effect of nonsynonymous coding variants on protein function was done with SIFT and PolyPhen scores. Both SIFT and PolyPhen output qualitative prediction scores (i.e. probably damaging/deleterious, possibly damaging/deleterious-low confidence, tolerated/benign). Noncoding variation scoring was assessed using CADD score, which is based on ranking a variant relative to all possible substitutions of the human genome. In addition, we explored the effect of significant loci on RNA and protein expression using the GTEx database[94] and the effect of significant genes using the Protein Atlas[95].

**Approximate conditional analyses of association**. In order to identify if the genome-wide significant loci of our original logistic meta-analyses were driven by one or more independent variants, we conducted approximate conditional and joint analyses of association (COJO) with the programme Genome-Wide Complex Trait Analysis (GCTA)[46]. We performed the analysis (–cojo-slct) using as input the summary statistics of our hair colour meta-analyses (fixed effects, FE) and the weighted average effect allele frequency from all studies. In addition, the programme requires a reference sample for computing LD correlations and, in the case of a meta-analysis, it is suggested to use one of the study's large samples[46]. Therefore, we ran the analysis twice: 1) using as a reference the sample genotyped with the Axiom UKBB array ($N = 4745$), and 2) using as a reference the sample genotyped with the GSA 24v1 + MDP ($N = 4480$), including in both cases only high imputation-score SNPs (INFO > 0.8). We assumed that variants farther than 10 Mb are in complete linkage equilibrium, hence we ran the programme on each chromosome separately to speed up the analyses and used the genome-wide significant p-value threshold of 1.67e-08.

**Statistical fine-mapping of significant loci**. We used the programme FINEMAP (version 1.4)[49] to identify candidate causal variants in the genome-wide associated loci across the genome for each binary phenotype. FINEMAP is based on a Bayesian framework, which uses summary statistics and LD correlations among variants to compute the posterior probabilities of causal variants, with a shotgun stochastic search algorithm[49]. Compared to other methods, FINEMAP allows a maximum of 20 causal variants per locus. To run the programme, we used as input the meta-analyses summary statistics of each binary phenotype, including the weighted average MAF among all studies, and an LD correlation matrix from one of the large samples in our study (Axiom UKBB array, $N = 4,745$). The LD correlation matrix was computed using LDStore (version 2.0), which considers

genotype probabilities[96]. We defined regions for fine-mapping as ±500 kb regions flanking the lead SNP, based on the genome-wide signals of association from the meta-analyses, and allowing a maximum number of 10 causal signals for each locus (i.e., a maximum of 10 credible sets). A credible set is comprised of SNPs that cumulatively reach a probability of at least 95%. The SNPs within a credible set are referred to as candidate causal variants and each of them has a corresponding posterior inclusion probability (PIP).

We filtered FINEMAP results by removing candidate causal variants with a $\log_{10}BF < 2$ from each of the 95% credible sets, where a $\log_{10}BF$ indicates considerable evidence of causality. We annotated the remaining SNPs using SNPnexus[51] to obtain information about the overlapping/nearest genes, overlapping regulatory elements and CADD scores. Annotation of gene expression on ENCODE, Roadmap Epigenomics and Ensembl Regulatory Build was restricted to melanocytes, keratinocytes and fibroblasts, which are the relevant cell types involved in hair pigmentation. Based on the combined evidence of fine-mapping and posterior annotation, we defined the candidate causal variants with strong evidence of causality (based on their $\log_{10}BF$ and annotation) as the most likely candidate causal variants. We computed LD correlations among the candidate causal SNP(s) on each locus using LDStore (version 2.0) and plotted the Posterior Inclusion Probability (PIP) and $\log_{10}BF$ results on R (version 3.5.1)[85] using ggplot2[89].

**Gene expression and methylation using cultured melanocyte data**. We conducted colocalization analyses of our GWAS meta-analyses signals using gene expression and methylation cis-QTL data from primary cultures of foreskin melanocytes, isolated from the foreskin of 106 newborn males[54,76]. Cis-QTLs were assessed for variants in the ±1 Mb region of each gene or CpG[54,76]. Foreskin melanocytes are currently the most adequate choice to study regulatory mechanisms involved in hair colour due to the shared pigmentation pathways in skin and hair. We used the programme hyprcoloc[53] to obtain the posterior probability of a variant being shared between the GWAS and the expression or methylation QTLs. We tested all the significant eQTL genes or meQTL probes within ±250 kb regions flanking the most significant GWAS SNP on each of the genome-wide regions of association (p-value ≤ 1.67e-08) from the logistic meta-analyses summary statistics (11 different loci across the three GWAS models). We used as LD reference the matrix obtained from the CanPath's Axiom UKBB Array (INFO score >0.3), computed on PLINK (version 1.9; –r square)[80,81]. We kept and report colocalized regions that reached a posterior probability ≥0.8, indicating high confidence of shared signal.

We performed three transcriptome-wide association studies (TWAS) by imputing the expression profile of the CanPath cohort using GWAS summary statistics and melanocyte RNA-seq expression data[54]. Using the programme FUSION[37], we used as LD reference the CanPath's Axiom UKBB genotyping array computed in binary PLINK format (version 1.9; –make-bed)[80,81]. As recommended by FUSION, we used the LDSC munge_sumstats.py script to check the GWAS summary statistics[90]. Before running the script, we filtered out SNPs with MAF < 0.01, SNPs with a genotyping missing rate >0.01 and SNPs that failed HWE test at significance threshold of 10e-07 using PLINK (version 1.9; –maf 0.01, –geno 0.01, –hwe 10e-7)[80,81]. We computed expression weights from our melanocyte RNA-seq data one gene at a time. Genes that failed quality control during a heritability check (using minimum heritability p-value of 0.01) were excluded from the further analyses, yielding a total of 3998 genes. We restricted the locus to 500 kb on either side of the gene boundary. We applied a significance cut-off to the final TWAS result of 4.17e-06 (i.e. 0.05/(3998 genes x 3 models tested)). Finally, we performed conditional analysis on FUSION (FUSION_post.process.R script) if more than one gene in a locus was significant, to identify if these were independent signals. We also ran a follow-up permutation test with a maximum of 100,000 permutations to assess if the random distribution of QTL effect sizes could yield a significant association by chance.

**Biological pathway gene set enrichment analysis**. We performed pathway enrichment analysis using the GENE2FUNC application available on the web-based programme FUMA[97], to annotate relevant gene sets in a biological context. FUMA tests if genes are overrepresented in any pre-defined Gene Ontology (GO) gene set, and the significance (p < 0.05) is adjusted for multiple testing per biological category separately, using the Bonferroni method. We used as input the gene IDs that overlapped candidate causal variants defined by FINEMAP for all models jointly (Supplementary Table 4), all genes as a background set (19,283 protein-coding genes), and a minimum overlap of two genes per gene set.

**Statistics and reproducibility**. Statistical analyses were performed for the GWAS and downstream analyses as described in the corresponding Methods section, including all parameters used to allow reproducibility.

**Reporting Summary**. Further information on research design is available in the Nature Research Reporting Summary linked to this article.

## Data availability

We provide the genome-wide (p ≤ 1.67e-8) and suggestive (p ≤ 1e-6) signals identified in the meta-analyses as a Supplementary Data (Supplementary Data 2–4). Further

information and requests for data published here should be directed to CanPath, which regulates the access to the data and biological materials (https://canpath.ca/). Melanocyte genotype data, RNA-seq expression data, and all meQTL association results are deposited in Genotypes and Phenotypes (dbGaP) under accession dbGaP: phs001500.v1.p1. The raw data of Illumina HumanMethylation450 BeadChips from 106 primary human melanocytes have been submitted to the Gene Expression Omnibus (GEO) database with the accession numbers: GSE101771 and GSE166069, respectively.

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

## Acknowledgements

The data used in this research were made available by CanPath—Canadian Partnership for Tomorrow's Health (formerly CPTP), CARTaGENE, Alberta's Tomorrow Project, Ontario Health Study, BC Generations Project and Atlantic PATH. The authors would like to thank all the participants of the Canadian Partnership for Tomorrow's Health. FLD is supported by the National Council for Science and Technology (CONACYT) in Mexico. MM was supported by a Mitacs Globalink Research Award (FR37903) and by Coordenação de Aperfeiçoamento de Pessoal de Nível Superior (CAPES) (88887.474324/2020-00). EJP received funding from the Natural Sciences and Engineering Research Council of Canada (NSERC Discovery Grant). RT, KF, MAK, JC, TZ, and KMB are supported by the Intramural Research Programme of the NIH, National Cancer Institute, Division of Cancer Epidemiology and Genetics; https://dceg.cancer.gov/); the content of this publication does not necessarily reflect the views or policies of the Department of Health and Human Services, nor does mention of trade names, commercial products, or organisations imply endorsement by the U.S. Government. Computations were performed on the GPC supercomputer at the SciNet HPC Consortium, Canada and at the UTM High-Performance Computing server at Mississauga, ON, Canada. This work also utilised the computational resources of the NIH HPC Biowulf cluster (http://hpc.nih.gov). SciNet is funded by: the Canada Foundation for Innovation under the auspices of Compute Canada; the Government of Ontario; Ontario Research Fund-Research Excellence; and the University of Toronto.

## Author contributions

E.J.P. and F.L.D. designed the study. F.L.D., M.M., and R.T. performed statistical analyses. F.L.D. wrote the draft of the manuscript. F.L.D., E.J.P., K.F., T.Z., M.A.K., J.C., and K.M.B. aided in the interpretation of the results and in the preparation of the final version of the manuscript.

## Competing interests

The authors declare no competing interests.
