## [Transparent Peer Review File · Communications Biology]

Reviewers' comments:

Reviewer #1 (Remarks to the Author):

Lona-Durazo et al. perform a GWAS of hair colour in a Canadian population cohort collected across multiple provinces, with a strong bias towards individuals recruited from Quebec, consisting of 12-13k individuals in total. They perform a series of genome-wide analyses in each of 5 different cohorts that comprise the larger study cohort, and follow these up with a suite of post-GWAS analyses, including fine-mapping, TWAS and colocalization with gene expression and methylation QTLs in ex vivo melanocytes. Generally speaking, the analyses are technically OK, and the authors provide a degree of utility by relating their genetic findings to a relevant cell type. The latter should be commended when other studies take a stock-in-trade approach of running analyses versus every tissue and cell type they can without considering what the relevant tissue or cell type is for the trait of interest. I have some concerns relating to technical points in the genome-wide analyses, and in particular it strikes me that some of the analysis decisions could be hugely simplified to make the analyses, and manuscript, clearer.

Major comments:

- 1) To determine the ancestral similarity of the CanPath cohort individuals, compared to a reference population, the authors perform a PCA for each set of samples according which array they were genotyped on. This makes it somewhat tricky to assess the relative degree of importance for each analysis. Instead, why not project all individuals into the same PC-space, or perform a joint-PCA across all individuals after accounting for the overlap or array probes/variants? This is particularly relevant for down-stream analyses where the authors decide to perform a separate GWAS for each subset of individuals, with different numbers of PCs included as adjustments for ancestry (which itself is not the most powerful way to account for genetic similarities across the study cohort). Given that each analysis adjust for a different amount ancestry and population genetic variance, how does this affect the model inference? E.g. are the odds ratios concordant across regions for each analysis?
- 2) Related to the above, having different numbers of PCs in each analysis can lead to disparities in the amount of population structure that is accounted for in the different analyses. A more powerful, and straight forward, approach would be to have a single harmonised cohort, which could be analysed with a linear mixed model, and adjust for the different array batches using a fixed-effect blocking factor.
- 3) The authors note that there is a difference in proportions of males and females reporting black hair colour, yet there is no mention of how the sex-chromosomes are handled in their analyses. Is the difference in black hair between sexes genetic or a difference in perception/reporting?
- 4) With the current analysis strategy, the authors perform, at my count, 15 separate GWAS. How is this additional multiple testing burden accounted for?
- 5) In the colocalization analyses, there are several ideas speculated that would be amenable to Mendelian Randomisation analysis to provide, at least a degree of, evidence in favour or against these hypotheses. For instance, lines 720-721, the authors note there is haplotype sharing between SLC24A4 variants for eQTLs and meQTLs. This is an ideal opportunity to test whether shared genetic signals for these molecular traits is causally related to hair colour, which would hugely strengthen the overall message of the manuscript. A similar situation arises for the meQTL and OCA2 expression.

Minor comments:

- 1) The abstract states "nearly 13,000 individuals", yet the introduction (line 77), states "more than 12,000" <- perhaps using the specific number would be clearer.

- 2) Supplementary Figure 5 – what do the colours denote? The legend appears to be truncated.
- 3) What criteria was used to declare individuals as outliers in PCA space, and how were the 81 removals distributed across provinces?
- 4) Was there any indication of systematic differences in imputation quality between the different arrays?
- 5) Why perform variant filtering on INFO score and MAF after performing GWAS? This should be an a priori decision, and thus performed before association testing. Doing so will also reduce the overall multiple testing burden.
- 6) I believe INFO scores are computed by the imputation programs, not SNPTEST itself.
- 7) Line 304: "...indicating high confidence of shared causality." Strictly colocalization tests for a shared genetic signal, not causality. Please consider revising this statement.
- 8) For the meta-analyses, what value of Q was considered significant. One assumes this is from a chi-squared test, if so, how many degrees of freedom? Please state the p-value and Q-statistic.
- 9) Line 493: "high-LD friends"- this is a very strange turn of phrase. Maybe avoid anthropomorphising genetic variants.
- 10) When comparing the logistic regression and linear regression results there are disparities for the synonymous SNPs at the MC1R locus. Is this any indication of the differences in false positive rates between the 2 analyses? /How does this impact the authors conclusions and their selection of candidate causal variants?
- 11) Line 522: Is rs12913832 a meQTL because the probe overlaps a SNP? Can the authors corroborate if this is a genuine meQTL, or if it is a technical artefact induced by the probe overlapping the polymorphic site? Usually probes overlapping SNPs and other variants would be removed during QC.
- 12) Lines 539-531. The authors note that the effect of variants in their GWAS and eQTL are the opposite of their expectation for EDNRB. How do the authors reconcile these results against their expectation, e.g. could EDNRB function through some other mechanism, such as melanosome formation?
- 13) The authors note that none of their epistasis results hold up to multiple testing. Is this due to a lack of power or sensitivity? For instance, can the authors detect any of the epistatic interactions between MC1R and ASIP variants as described in Morgan et al as a potential sanity-check/control?
- 14) Lines 587-595: Overlapping SNPs between traits is not an indication of shared genetic signal. The authors should perform a formal colocalization analysis between traits to establish evidence of genetic sharing between these traits.
- 15) Lines 699-700: This seems somewhat speculative. How would differences in CDK10 expression lead to changes in hair colour? The authors should discuss the possibility of synthetic association here given that a common variant may tag multiple rare haplotypes giving the appearance of an association. Comparing D' for these CDK10 locus variants with MC1R non-synonymous SNPs might give an indication if these variants are shared across multiple MC1R red hair haplotypes.
- 16) Lines 723-726: Why do the authors specifically focus on DNA looping as a mechanism of action for a CpG region? What evidence do the authors have to support this?

17) Lines 726-731: This seems like a testable hypothesis, though perhaps beyond the scope of this manuscript. Is there public melanocyte data to back up whether such a change occurs between newborn and adult melanocytes (e.g. ENCODE/GTeX/Roadmap Epigenome)?

18) Lines 737-741: It's not clear how the expression of genes in whole blood (a mixture of cell types) relates to skin cancer risk and pigmentation. This requires clarification.

Reviewer #2 (Remarks to the Author):

Lona-Durazo and co-authors present a comprehensive study of the genetics of hair colour. While this is smaller than some recent studies, it does a good job of thoroughly integrating the data. I think all the conclusions are well supported, though some of the main findings are not novel. The methods section is comprehensive, though sometimes it reads a little too much like the authors are simply repeating the default parameters of some of the software used. In the discussion there are passages relating to the biological function of the candidate genes at those loci; but I think after this biology, it would be useful to have a concluding paragraph that ties this together, to set out the importance of these findings.

A few specific points:

The use of the lambda values to correct for population stratification is slightly out of data. It would be better to investigate the effects of population structure on the data using LD score regression, and specifically the intercept.

I'm not convinced about including loci at suggestive levels of significance without good reasons to do so. This also applies to the COJO analysis, where I notice that $5e-6$ is used.

Foreskin might be the best possible tissue to use for this study, but I think that a one sentence justification for readers less familiar with hair colour would be appreciated.

I'm quite intrigued by the gender imbalance, particularly for black hair, do the authors have any idea why this might be, and to they think it represents any features of the data collection that might be biasing their results?

Line 250, talks about eight loci, but the numbers in the brackets that follow add up to seven.

I'm a fan of supplementary figures 3 and 4, but I think they would be more informative, if there was a second panel with the same data, but coloured to indicate the hair colour of the participants.

In Supplementary Figure 7, am I correct that people in Ontario are more closely related to people in British Columbia, Alberta and the Atlantic provinces than they are to other people in Ontario? I know nothing about Canadian demography, but this seems bit odd.

This is purely a stylistic point, but for some of the supplementary tables, it might make sense to have column labels that are more easily understood, rather than to have a separate sheet with the definitions.

Reviewer #3 (Remarks to the Author):

Here the authors perform a genome-wide association study of several hair colour traits in a Canadian

cohort, substantially confirming previous results. They further fine map the identified loci and to understand the role of the putatively causal SNPs they use colocalization with eQTLs and meQTLs coming from a melanocyte dataset.

Overall, I believe the results of the authors especially since all of them have been previously reported, so in this respect, I do not feel that at least their main claims are due to chance.

There are several issues however with some analytical choices and the way the data is presented.

Starting from the latter, the paper is too long, I appreciate the detail but the text needs to be reduced to be readable.

For example there is a whole paragraph on the results from the epistasis analysis which fundamentally gives no results. This in itself is fine but there are too many details that make reading it quite difficult. I think that it could be summarised much better and possibly move some of the details to a supplementary paragraph.

I think it is possible that the authors may not have worked in the GWAS field before and thus the language is unusual.

I am not saying it is wrong but it is quite hard to read and the paper could use an improvement in this respect.

Also figure 1 is missing the legend and it looks like it is a screenshot or a photo of two papers one next to the other. Clearly, it is not suited for a scientific paper as it is.

Looking at the paper from the analytical point of view there I have a some issues with the way the GWAS was conducted. In particular I do not think that the PCs used by the authors are enough to correct for population stratification especially when it comes to hair colour genes which are known to be particularly variant between different populations. They should be using at least 10 if not 20 regardless of the association with the phenotype or how much genetic variance they explain. This becomes obvious when looking at the Manhattan plot for the "red hair" trait which shows several SNPs which look like they are artefacts due to population structure.

Also I would suggest that they use an association software which uses mixed models to account for cryptic relationships within their samples as options exist for both quantitative and binary traits. These are not to be considered alternative but should be used in combination as while the PCs estimated with the other populations capture the population level stratification, the mixed model will account for the more subtle relatedness.

I do not get why the authors run many different population structure analysis on their samples like ADMIXTURE and CHROMOPAINTING and then do not use the results of these analyses for their GWAS. What is the point of these results within this paper?

One option would be to use the results from the CHROMOPAINTING as covariate in the analysis as done here : <https://www.nature.com/articles/s41467-018-08219-1>. The authors instead use only two principal component. So either these results are removed from the paper or they need to be functional to the goal which is to finemap hair colour genes.

So although I think the paper is interesting and that the results are likely to be correct, I think it needs a serious revisiting of the main analyses and of the way they are presented, especially everything that is included in the paper. I realise the authors have put a lot of work in this study and they want to share it with the world, but it also needs to be readable and understandable by the reader.

We would like to thank the three reviewers for the very helpful comments that have led to an improved version of our manuscript. We have addressed point by point each of the comments below.

Reviewer #1 (Remarks to the Author):

Lona-Durazo et al. perform a GWAS of hair colour in a Canadian population cohort collected across multiple provinces, with a strong bias towards individuals recruited from Quebec, consisting of 12-13k individuals in total. They perform a series of genome-wide analyses in each of 5 different cohorts that comprise the larger study cohort, and follow these up with a suite of post-GWAS analyses, including fine-mapping, TWAS and colocalization with gene expression and methylation QTLs in ex vivo melanocytes. Generally speaking, the analyses are technically OK, and the authors provide a degree of utility by relating their genetic findings to a relevant cell type. The latter should be commended when other studies take a stock-in-trade approach of running analyses versus every tissue and cell type they can without considering what the relevant tissue or cell type is for the trait of interest. I have some concerns relating to technical points in the genome-wide analyses, and in particular it strikes me that some of the analysis decisions could be hugely simplified to make the analyses, and manuscript, clearer.

Major comments:

1) To determine the ancestral similarity of the CanPath cohort individuals, compared to a reference population, the authors perform a PCA for each set of samples according which array they were genotyped on. This makes it somewhat tricky to assess the relative degree of importance for each analysis. Instead, why not project all individuals into the same PC-space, or perform a joint-PCA across all individuals after accounting for the overlap or array probes/variants? This is particularly relevant for down-stream analyses where the authors decide to perform a separate GWAS for each subset of individuals, with different numbers of PCs included as adjustments for ancestry (which itself is not the most powerful way to account for genetic similarities across the study cohort). Given that each analysis adjust for a different amount ancestry and population genetic variance, how does this affect the model inference? E.g. are the odds ratios concordant across regions for each analysis?

We have revised our GWAS approach, and we have implemented logistic mixed models, including 10 PCs as fixed effects, independent of the genetic variance explained by the PCs. Additionally, we have included two heterogeneity measures in the meta-analysis (Cochran's Q and I^2), which indicate if the effect sizes (i.e. $\log(\text{odds ratio})$) of each of the studies are different. We have also included as Supplementary Figure 9 forest plots of the top genome-wide SNPs in each of the studies and in the meta-analyses.

2) Related to the above, having different numbers of PCs in each analysis can lead to disparities in the amount of population structure that is accounted for in the different analyses. A more powerful, and straight forward, approach would be to have a single harmonised cohort, which could be analysed with a linear mixed model, and adjust for the different array batches using a fixed-effect blocking factor.

As described above, we have revised our GWAS approach and have repeated all the analyses using logistic mixed models including 10 PCs as fixed effects for each sample. It is important to note that the

CanPath cohorts were genotyped with very different arrays, and this has an effect on the imputation of untyped genetic markers and therefore the possibility of doing the analysis as a single harmonised cohort. We opted to perform a meta-analysis using the summary results, including only markers with good imputation scores, as is typically done in this type of studies.

3) The authors note that there is a difference in proportions of males and females reporting black hair colour, yet there is no mention of how the sex-chromosomes are handled in their analyses. Is the difference in black hair between sexes genetic or a difference in perception/reporting?

Sex chromosomes data was not available for this study. Although we cannot discard the possibility of the effect being due to a bias in the hair colour self-report, we note that previous studies have also found a similar pattern, which suggests a genetic basis for the sex differences. Aside from the possible effect that sex chromosomes may have, it is also possible to evaluate SNP-by-sex statistical interactions. However, by exploring this approach across our genome-wide significant signals, we did not detect any significant SNP-by-sex interactions.

4) With the current analysis strategy, the authors perform, at my count, 15 separate GWAS. How is this additional multiple testing burden accounted for?

We have included a more stringent p-value threshold of $1.67e-8$ to account for the three models tested in our study. The purpose of a meta-analysis is to combine two or more studies into a single estimate to increase the power of the analysis, therefore there is no inherent multiple testing burden related to the number of studies included.

5) In the colocalization analyses, there are several ideas speculated that would be amenable to Mendelian Randomisation analysis to provide, at least a degree of, evidence in favour or against these hypotheses. For instance, lines 720-721, the authors note there is haplotype sharing between SLC24A4 variants for eQTLs and meQTLs. This is an ideal opportunity to test whether shared genetic signals for these molecular traits is causally related to hair colour, which would hugely strengthen the overall message of the manuscript. A similar situation arises for the meQTL and OCA2 expression.

We thank the reviewer for this interesting suggestion. These analyses are beyond the scope of the present paper. We are aiming to study the causal effect of the intermediate phenotypes and pigmentation in a separate paper, given that we are also working with eye colour data from the same cohort and this would also allow us to explore these issues for both pigimentary traits.

Minor comments:

1) The abstract states “nearly 13,000 individuals”, yet the introduction (line 77), states “more than 12,000” <- perhaps using the specific number would be clearer.

We have changed this phrase in the abstract and in the introduction to precise the total number. In the abstract (lines 21-22), it now reads as: “Here, we conducted GWAS meta-analyses of hair colour in a Canadian cohort of 12,741 individuals of European ancestry.” Similarly, in the introduction (lines 71-73) it now reads as: “In this study, we conducted a meta-analysis of genome-wide association studies

including 12,741 Canadian participants of European-related ancestry from the Canadian Partnership for Tomorrow's Health.”

2) Supplementary Figure 5 – what do the colours denote? The legend appears to be truncated.

We have removed this figure for simplification of the manuscript.

3) What criteria was used to declare individuals as outliers in PCA space, and how were the 81 removals distributed across provinces?

We have specified in the Methods section that we removed PCA outliers by inspecting the first three principal components, and we have indicated the distribution of the outliers according to the different provinces. The paragraph now reads as (lines 459-461): “Finally, we performed a PCA with the full 1KGP Phase 3 samples and removed individual outliers that did not cluster within the European sample of the 1KGP by inspecting the first three principal components (total PCA outliers across genotyping arrays = 81). Amongst the outliers, 63 individuals are from Quebec, 8 from British Columbia, 5 from the Atlantic Provinces, 5 from Alberta and none from Ontario.”

4) Was there any indication of systematic differences in imputation quality between the different arrays?

As described above, the CanPath cohorts were genotyped with different arrays: Primarily the Axiom 2.0 UK Biobank, Infinium Omni 2.5 and different versions of the Infinium Global Screening Array. These arrays have different numbers of markers and different imputation backbones. However, it is important to note that only variants with INFO scores > 0.3 were included in the Genome-wide Association Analyses. Therefore, poorly imputed variants were excluded from the analyses. The final number of markers included in the GWAS for each genotyping array were: (i) Axiom 2.0 UK Biobank = 6,880,138, (ii) GSA 24v1 = 6,185,935, (iii) GSA 24v2+MDP = 6,214,597, (iv) GSA 24v1+MDP = 6,204,261, and (v) Omni 2.5 = 7,391,256. We have included this information in the manuscript, it now reads as (lines 481-483): “The final number of markers included in the GWAS for each genotyping array were: (i) Axiom 2.0 UK Biobank = 6,880,138, (ii) GSA 24v1 = 6,185,935, (iii) GSA 24v2+MDP = 6,214,597, (iv) GSA 24v1+MDP = 6,204,261, and (v) Omni 2.5 = 7,391,256.”

5) Why perform variant filtering on INFO score and MAF after performing GWAS? This should be an a priori decision, and thus performed before association testing. Doing so will also reduce the overall multiple testing burden.

While redoing our analyses, we have performed the INFO score and MAF filtering before conducting the GWAS and stated it in the Methods (lines 477-480).

6) I believe INFO scores are computed by the imputation programs, not SNPTEST itself.

We have clarified this in the Methods section (lines 478-479): “Briefly, the INFO score is a measure of the imputation certainty across samples, in which INFO = 1 indicates complete certainty, whereas INFO = 0 indicates that the genotype probabilities for each sample are completely uncertain.”

7) Line 304: "...indicating high confidence of shared causality." Strictly colocalization tests for a shared genetic signal, not causality. Please consider revising this statement.

We have revised this statement, which now reads as (lines 600-601): "We kept colocalized regions that reached a posterior probability ≥ 0.8 , indicating high confidence of shared signal."

8) For the meta-analyses, what value of Q was considered significant. One assumes this is from a chi-squared test, if so, how many degrees of freedom? Please state the p-value and Q-statistic.

We have clarified this in the Methods section (lines 517-519). It now reads as: "We considered a SNP as heterogeneous across studies at an alpha level of 0.05 and $K-1$ degrees of freedom, where K is the number of studies included in the meta-analysis. Similarly, values of $I^2 > 50\%$ are considered to represent notable heterogeneity." We have also included the Cochran's statistic as well as the p-values throughout the results.

9) Line 493: "high-LD friends"- this is a very strange turn of phrase. Maybe avoid anthropomorphising genetic variants.

We have removed this term from the manuscript.

10) When comparing the logistic regression and linear regression results there are disparities for the synonymous SNPs at the MC1R locus. Is this any indication of the differences in false positive rates between the 2 analyses? /How does this impact the authors conclusions and their selection of candidate causal variants?

We based our downstream analyses, including fine-mapping, in the logistic mixed models, not in the linear model. Given that we did not consider the red hair colour category in the linear model, this model would capture SNPs in the MC1R locus involved in eumelanogenesis and not pheomelanogenesis (i.e. similar to the blonde vs. brown + black hair colour).

11) Line 522: Is rs12913832 a meQTL because the probe overlaps a SNP? Can the authors corroborate if this is a genuine meQTL, or if it is a technical artefact induced by the probe overlapping the polymorphic site? Usually probes overlapping SNPs and other variants would be removed during QC.

The probe does not overlap the SNP, instead the probes are located upstream and downstream of the rs12913832 polymorphism. We have clarified this sentence by mentioning the CpG probes' IDs. It now reads as (lines 226-227): "On the *OCA2* locus, rs12913832 is a meQTL for a CpG tagged by probes located upstream and downstream of the SNP (cg05271345, cg25622125 and cg27374167)."

12) Lines 539-531. The authors note that the effect of variants in their GWAS and eQTL are the opposite of their expectation for *EDNRB*. How do the authors reconcile these results against their expectation, e.g. could *EDNRB* function through some other mechanism, such as melanosome formation?

We have expanded our explanation regarding the unexpected direction of effect of *EDNRB*. It now reads as follows (lines 245-251): "In contrast, the increased expression of *EDNRB* was associated with blonde hair colour, which was not identified through colocalization analyses. In this case, the direction of effect in both the eQTL and GWAS is negative, which is the opposite of what was expected, given that the protein encoded by *EDNRB* is involved in melanocyte development and it induces melanocyte proliferation⁷⁵. This

discrepancy could be due to different direction of effects in skin and hair melanocytes, similar to the inverse effect of *IRF4* effect on hair and skin pigmentation⁷⁶. However, further investigation of the *EDNRB* expression patterns in the hair bulb is needed to provide a clear explanation.”

13) The authors note that none of their epistasis results hold up to multiple testing. Is this due to a lack of power or sensitivity? For instance, can the authors detect any of the epistatic interactions between *MC1R* and *ASIP* variants as described in Morgan et al as a potential sanity-check/control?

We have removed this section from the manuscript for simplification, given that it does not substantially contribute to the main results.

14) Lines 587-595: Overlapping SNPs between traits is not an indication of shared genetic signal. The authors should perform a formal colocalization analysis between traits to establish evidence of genetic sharing between these traits.

We have clarified that an overlap does not imply a shared signal by adding the following sentence (lines 285-287): “We note that an overlap of significant signals is not indicative of a shared causal signal between traits, therefore the biological relevance of these loci on skin cancer should be functionally investigated.”. It is not possible to conduct a colocalization analysis, given that we use here the genome-wide significant SNPs identified across multiple studies available in the GWAS Catalog.

15) Lines 699-700: This seems somewhat speculative. How would differences in *CDK10* expression lead to changes in hair colour? The authors should discuss the possibility of synthetic association here given that a common variant may tag multiple rare haplotypes giving the appearance of an association. Comparing D' for these *CDK10* locus variants with *MC1R* non-synonymous SNPs might give an indication if these variants are shared across multiple *MC1R* red hair haplotypes.

We do not think that the expression of *CDK10* is involved in hair colour variation. In fact, even though our TWAS for red hair colour highlighted the expression of *CDK10*, we have noted that this gene is not involved in hair colour variation (lines 258-259). In contrast, we suggest that there may be non-coding variants in the *MC1R* vicinity, such as the meQTLs within *CDK10* that also contribute to eumelanin variation in hair by regulating the expression of *MC1R*, on top of the known missense SNPs (lines 360-362).

16) Lines 723-726: Why do the authors specifically focus on DNA looping as a mechanism of action for a CpG region? What evidence do the authors have to support this?

We have clarified and specified the sentence and we have added a reference. It now reads as (lines 382-384): “These loci do not harbour colocalizing eQTLs, which suggests that other mechanisms may be involved, such as *trans*-QTLs⁶⁴, which were not considered in the current analysis.”

17) Lines 726-731: This seems like a testable hypothesis, though perhaps beyond the scope of this manuscript. Is there public melanocyte data to back up whether such a change occurs between newborn and adult melanocytes (e.g. ENCODE/GTEx/Roadmap Epigenome)?

The ENCODE Project includes foreskin melanocyte data for both histone modification markers (H3K27me3 and H3K27ac) in newborns, whereas Roadmap Epigenomics only has skin-related data for fetal tissues. Additionally, there is no eQTL data for melanocytes in the ENCODE Project. The GTEx has information for

adult skin tissue as a whole (which includes several cell-types) and we have checked if the two candidate colocalized SNPs (rs258322 and rs7773997) are eQTLs in skin tissue, but did not find support for it. We have added this within the paragraph, which now reads as follows (lines 384-391):

“Alternatively, it is possible that some CpG islands capture the status of poised enhancers (i.e. enhancers in a *latent* state), which may not yet have any influence on gene expression in actively growing melanocytes. This is a possible scenario, given that the melanocytes used in the QTL analyses were from newborns. However, none of the candidate colocalized SNPs in these loci (rs258322 and rs7773997) are eQTLs of *MC1R* and *IRF4*, respectively, in adult skin tissue based on the GTEx Project. Experimental histone modification marker assays may provide support for the alternative hypothesis, as it is known that poised enhancers lose H3K27me3 and acquire acetylation at the same amino acid residue upon activation⁹⁷”.

18) Lines 737-741: It’s not clear how the expression of genes in whole blood (a mixture of cell types) relates to skin cancer risk and pigmentation. This requires clarification.

We have compared our study to that of Bonilla et al. (2020), given that it is the first study characterizing regulatory mechanisms, such as methylation, in pigmentation-related phenotypes. However, we agree with the reviewer that whole blood is not the best proxy, and we have indicated this in our discussion (lines 404-406): “The differences may lie in the fact that we used cultured melanocytes, which provide a cell-specific expression and methylation profile, best suited for the traits being tested.”.

Reviewer #2 (Remarks to the Author):

Lona-Durazo and co-authors present a comprehensive study of the genetics of hair colour. While this is smaller than some recent studies, it does a good job of thoroughly integrating the data. I think all the conclusions are well supported, though some of the main findings are not novel. The methods section is comprehensive, though sometimes it reads a little too much like the authors are simply repeating the default parameters of some of the software used. In the discussion there are passages relating to the biological function of the candidate genes at those loci; but I think after this biology, it would be useful to have a concluding paragraph that ties this together, to set out the importance of these findings.

A few specific points:

1) The use of the lambda values to correct for population stratification is slightly out of data. It would be better to investigate the effects of population structure on the data using LD score regression, and specifically the intercept.

We have computed LD Score Regression and we have incorporated into our results the LD score intercept for the meta-analyses, which indicates no residual population stratification (lines 114-116): “Additionally, the LD Score regression intercepts computed on LDSC (version 1.0.1) were 1.001, 0.994 and 0.999 for the three models tested, respectively, indicating no residual confounding bias.”

2) I'm not convinced about including loci at suggestive levels of significance without good reasons to do so. This also applies to the COJO analysis, where I notice that 5e-6 is used.

We have restricted our analyses to include only genome-wide significant signals, accounting also for the three models used in our manuscript (p -value threshold = $1e-67$).

3) Foreskin might be the best possible tissue to use for this study, but I think that a one sentence justification for readers less familiar with hair colour would be appreciated.

We have added a sentence in the Methods section, which now reads as follows (lines 593-594):
“Foreskin melanocytes are currently the most adequate choice to study regulatory mechanisms involved in hair colour due to the shared pigmentation pathways in skin and hair.”.

4) I'm quite intrigued by the gender imbalance, particularly for black hair, do the authors have any idea why this might be, and to they think it represents any features of the data collection that might be biasing their results?

Although we cannot discard the possibility of the effect being due to a bias in the hair colour self-report, we note that previous studies have also found a similar pattern, which suggests a genetic basis for the sex differences (e.g. Morgan et al. 2018). Aside from the possible effect that sex chromosomes may have, it is also possible to evaluate SNP-by-sex statistical interactions. However, by exploring this approach across our genome-wide significant signals, we did not detect any significant SNP-by-sex interactions and have not included this section in the manuscript.

5) Line 250, talks about eight loci, but the numbers in the brackets that follow add up to seven.

We have removed this section of the manuscript, given that we consider only genome-wide significant signals (p-value threshold = $1.67e-8$) throughout, and we did not identify novel loci.

6) I'm a fan of supplementary figures 3 and 4, but I think they would be more informative, if there was a second panel with the same data, but coloured to indicate the hair colour of the participants.

We have updated the PCA plots (Supplementary Figures 2 and 3) by colouring the CanPath samples by hair colour categories.

7) In Supplementary Figure 7, am I correct that people in Ontario are more closely related to people in British Columbia, Alberta and the Atlantic provinces than they are to other people in Ontario? I know nothing about Canadian demography, but this seems bit odd.

We have removed this section of the results for simplification of the manuscript.

8) This is purely a stylistic point, but for some of the supplementary tables, it might make sense to have column labels that a more easily understood, rather than to have a separate sheet with the definitions.

We have updated the column codes of the Supplementary files and have removed the column codes sheet.

Reviewer #3 (Remarks to the Author):

Here the authors perform a genome-wide association study of several hair colour traits in a Canadian cohort, substantially confirming previous results. They further fine map the identified loci and to understand the role of the putatively causal SNPs they use colocalization with eQTLs and meQTLs

coming from a melanocyte dataset.

Overall, I believe the results of the authors especially since all of them have been previously reported, so in this respect, I do not feel that at least their main claims are due to chance.

There are several issues however with some analytical choices and the way the data is presented.

1) Starting from the latter, the paper is too long, I appreciate the detail but the text needs to be reduced to be readable.

For example there is a whole paragraph on the results from the epistasis analysis which fundamentally gives no results. This in itself is fine but there are too many details that make reading it quite difficult. I think that it could be summarised much better and possibly move some of the details to a supplementary paragraph.

We have revised our manuscript and we have thus reduced its length by removing those sections which do not contribute substantially to our main results, to make it more readable.

2) I think it is possible that the authors may not have worked in the GWAS field before and thus the language is unusual. I am not saying it is wrong but it is quite hard to read and the paper could use an improvement in this respect.

We would like to clarify that some of the authors of the manuscript have been involved in many GWAS studies, not only of pigimentary traits (skin, hair and eye colour), but also many other traits of biomedical importance, including type 2 diabetes, lipids, and anthropometric traits.

3) Also figure 1 is missing the legend and it looks like it is a screenshot or a photo of two papers one next to the other. Clearly, it is not suited for a scientific paper as it is.

We have updated Figure 1 of the manuscript to increase the quality of the plots.

4) Looking at the paper from the analytical point of view there I have a some issues with the way the GWAS was conducted. In particular I do not think that the PCs used by the authors are enough to correct for population stratification especially when it comes to hair colour genes which are known to be particularly variant between different populations. They should be using at least 10 if not 20 regardless of the association with the phenotype or how much genetic variance they explain. This becomes obvious when looking at the Manhattan plot for the "red hair" trait which shows several SNPS which look like they are artefacts due to population structure.

Also I would suggest that they use an association software which uses mixed models to account for cryptic relationships within their samples as options exist for both quantitative and binary traits. These are not to be considered alternative but should be used in combination as while the PCs estimated with the other populations capture the population level stratification, the mixed model will account for the more subtle relatedness.

We have revised the GWAS approach used in the manuscript, based on the reviewer's comments. We have run our GWAS using logistic mixed models and included 10 PCs, independently of the genetic variation explained by each of the PCs. The results are almost identical to those observed in the previous

analyses.

5) I do not get why the authors run many different population structure analysis on their samples like ADMIXTURE and CHROMOPAINTING and then do not use the results of these analyses for their GWAS. What is the point of these results within this paper?

One option would be to use the results from the CHROMOPAINTING as covariate in the analysis as done here : <https://www.nature.com/articles/s41467-018-08219-1>. The authors instead use only two principal component. So either these results are removed from the paper or they need to be functional to the goal which is to finemap hair colour genes.

In order to simplify the manuscript, we have removed the ancestry analyses that do not contribute substantially to the main results of the paper.

So although I think the paper is interesting and that the results are likely to be correct, I think it needs a serious revisiting of the main analyses and of the way they are presented, especially everything that is included in the paper. I realise the authors have put a lot of work in this study and they want to share it with the world, but it also needs to be readable and understandable by the reader.

REVIEWERS' COMMENTS:

Reviewer #1 (Remarks to the Author):

The authors should be commended for substantially revising their manuscript and greatly increasing the clarity of writing.

My only comment is that the authors should pay careful attention to the distinction between CpG dinucleotides (what the methylation array probes target) and CpG islands (a computationally defined genomic interval), which are not the same. These names are sometimes used the wrong way around in the manuscript.

Reviewer #2 (Remarks to the Author):

I thank the authors for clear responses to my questions, and I think that the the paper has substantially improved. I do however have two remaining areas of comment.

The first relates to the response to my comment about people in Ontario and how they appeared to be genetically related to people in other provinces. I thought that this was a slightly odd results, but the author's response is to removed it from the paper. This isn't quite enough for me - the data being used in this study is still the same. If there was a mistake in that analysis, and it has been corrected; fine. But if that structure still exists in the data, then I would like an explanation.

The second set of points are about how the eQTL analysis is described. It is a little hard to follow exactly what was being tested. If, as it appears, that the authors were only testing the eQTL with their candidate genes, they should make this much clearer. Indeed, it would be better if the authors were to report any gene for which there was evidence from the eQTL data (even if it wasn't one of their candidate genes), to allow readers as full a picture as possible. I'm also a bit confused by the comment that the MC1R variants wouldn't be eQTLs because they are non-synonymous; that might be the case here (but I want more information); but it isn't a general rule of eQTLs. Indeed cis-eQTLs are very much enriched for non-synonymous variants (see [dx.doi.org/10.1016/j.celrep.2017.05.018](https://doi.org/10.1016/j.celrep.2017.05.018)).

We would like to thank the reviewers for the comments and feedback on the revised version of our manuscript. We have addressed each of the comments below, and highlighted where changes have been made within the manuscript.

REVIEWERS' COMMENTS:

Reviewer #1 (Remarks to the Author):

The authors should be commended for substantially revising their manuscript and greatly increasing the clarity of writing.

My only comment is that the authors should pay careful attention to the distinction between CpG dinucleotides (what the methylation array probes target) and CpG islands (a computationally defined genomic interval), which are not the same. These names are sometimes used the wrong way around in the manuscript.

We have revised our manuscript according to the reviewer's comment. We have specifically improved the clarity of the Results and Discussion sections regarding CpGs, which now read as follows:

Lines 223-227: "In the case of *SLC24A4*, the SNP rs8022442 is a meQTL for CpG probes, cg11086312 and cg10004481. On the *OCA2* locus, rs12913832 is a meQTL for CpG probes located upstream and downstream of the SNP within *HERC2* (cg05271345, cg25622125 and cg27374167). Furthermore, the SNP rs35391 is a meQTL for the CpG probes in the first exon of *SLC45A2* region (cg14189614 and cg04302388)."

Lines 232-236: "Nonetheless, we did identify colocalizing meQTLs (rs258322) for blonde hair colour, associated with the CpG methylation near *MC1R*: on or near *CDK10*, *GAS8* and *DPEP1* (cg05714116, cg06907930 and cg00996377), which may point at an independent regulatory region associated with blonde hair colour, apart from the known missense SNP within *MC1R* (rs1805005)³⁹."

On Table 1, we have improved the legend explaining the Methylation Annotation: "The Gene/Methylation Annotation indicates the location of CpG probes with respect to the nearest gene, as well as relative to CpG island."

Lines 336-337: "However, we cannot be certain of a correlation between the meQTL target CpG and *OCA2* expression, given the current evidence."

Lines 379-384: "Our colocalization results highlighted meQTLs for blonde hair colour, associated with the methylation of CpGs near known pigmentation genes (i.e. *MC1R*, *IRF4*). These loci do not harbour colocalizing eQTLs, which suggests that other mechanisms may be involved, such as *trans*-QTLs⁷⁶, which were not considered in the current analysis. Alternatively, it is possible that some CpGs capture the status of poised enhancers (i.e. enhancers in a *latent* state), which may not yet have any influence on gene expression in actively growing melanocytes."

Reviewer #2 (Remarks to the Author):

I thank the authors for clear responses to my questions, and I think that the the paper has

substantially improved. I do however have two remaining areas of comment.

The first relates to the response to my comment about people in Ontario and how they appeared to be genetically related to people in other provinces. I thought that this was a slightly odd results, but the author's response is to remove it from the paper. This isn't quite enough for me - the data being used in this study is still the same. If there was a mistake in that analysis, and it has been corrected; fine. But if that structure still exists in the data, then I would like an explanation.

The reviewer is probably referring to Supplementary Figure 7 in our original version of the paper. This figure reported the average IBD sharing observed both within and between provinces. This figure indicated that the average IBD sharing observed in the Ontario sample is slightly lower than the average sharing observed between Ontario and other provinces (Alberta, Atlantic Provinces and British Columbia). We would like to point out that the average sharing within the sample from British Columbia is also slightly lower than what is observed between British Columbia and Alberta. In all cases, the differences in IBD sharing are very small. These analyses based on IBD sharing reflect recent common ancestry in the individuals of the CanPath samples. The results presented in Supplementary Figure 7 were based on analysis using the Hap-IBD program. We also analyzed the data using an alternative program (i.e. Refined IBD), and we observed similar trends: the average IBD sharing within the province of Ontario was slightly lower than the IBD sharing observed between Ontario and Alberta, Atlantic Provinces and British Columbia. Also, in agreement with the Hap-IBD results, IBD sharing within British Columbia was slightly lower than IBD sharing between British Columbia and Alberta. There is very clear concordance in the results using both methods. We have repeated the IBD analysis with the program Hap-IBD after phasing again the data using two different approaches (i.e. Eagle and Shapeit4), with consistent results.

There are several explanations for the IBD results observed in the CanPath samples, which reflect recent common ancestry trends. First, there is substantial interprovincial migration in Canada. This information can be accessed at: <https://www150.statcan.gc.ca/t1/tbl1/en/tv.action?pid=1710002201>. Ontario, as the largest Province in Canada and one of the major economic hubs of the country, attracts migrants from many other provinces (including Alberta, the Atlantic Provinces and British Columbia). There is also substantial migration from Ontario to other Canadian Provinces, primarily Alberta and British Columbia. The data also indicates very substantial migration between Alberta and British Columbia. This interprovincial migration patterns may explain in part the IBD results we are observing in the CanPath sample. It is also important to mention that there has been very substantial migration from other countries to Canada (more than 200,000 migrants per year since 2004, and more than 250,000 in the last five years, <https://www.statista.com/statistics/443063/number-of-immigrants-in-canada/>), with Ontario, British Columbia, Alberta and Quebec receiving the bulk of the immigrants, and this will also influence the IBD sharing patterns observed in the CanPath samples. The diverse origins of the individuals of European descent observed in the CanPath data is apparent in the PC plots that we provided in the original manuscript (Supplementary Figure 4). Although most of the samples cluster with the 1KG GBR and CEU samples, there are many samples that cluster with other European regions (e.g. IBS, TSI, or even Finland). Interestingly, the majority of the samples from Alberta, British Columbia and Atlantic Canada cluster together in the plot, and have a substantial overlap, emphasizing again that there are no substantial genetic differences between those provinces (although the Ontario samples show more dispersion in the plots, implying more diverse origins). This contrasts with the sample of Quebec, which in agreement with the demographic origins of this population (e.g. substantial French ancestry), is located closer to the Spanish and Italian 1KG samples than any of the other provinces. Incidentally, the PC representation (axes 1/3) also points to a founder effect in Quebec, as many of the

Quebec samples tend to occupy extreme positions on PC 3. This founder effect has been extensively reported in the scientific literature (e.g. <https://www.annualreviews.org/doi/abs/10.1146/annurev.genom.2.1.69?journalCode=genom>). Consistent with this, we showed that IBD sharing in Quebec was higher than in any other province (this is observed using both Hap-IBD or Refined IBD). In fact, based on this population history, we were expecting to see much higher IBD sharing in Quebec than what was observed in the CanPath sample. This may be due to the sampling strategy used in the CartaGene cohort (e.g. the cohort from Quebec in CanPath), where the bulk of the sampling happened in Montreal, which is a cosmopolitan city with people from many diverse origins.

In summary, our IBD analyses in the CanPath samples were carried out with two different programs with very consistent results, and we have repeated the analysis after phasing obtaining similar trends. We believe that the results of the IBD analyses are not a mistake of our analyses, and reflect the particular demographic composition of the samples of the CanPath cohorts. In any case, it is important to note that this does not influence in any way the results of our GWAS analyses, as we have implemented mixed models that take into account population structure and potential relatedness in the samples; as described in the manuscript, there is no evidence of residual population structure based on the QQ plots or LD Score regression intercepts. We eliminated this section of the paper because several reviewers indicated that the manuscript was very long and we felt it was not necessary to provide a detailed demographic description of the CanPath samples, as this does not influence the results provided in the paper.

The second set of points are about how the eQTL analysis is described. It is a little hard to follow exactly what was being tested. If, as it appears, that the authors were only testing the eQTL with their candidate genes, they should make this much clearer. Indeed, it would be better if the authors were to report any gene for which there was evidence from the eQTL data (even if it wasn't one of their candidate genes), to allow readers as full a picture as possible.

We understand that the reviewer is asking if we tested colocalization only for a select candidate genes in the GWAS region. In fact, we performed the colocalization analyses by testing all the significant *cis*-eQTL genes and *cis*-meQTL probes within ± 250 kb of the GWAS lead SNP, and we only present those that show evidence of colocalization (posterior probability > 0.8). We have made this clearer in the Methods section of the manuscript:

Lines 591-593: “We conducted colocalization analyses of our GWAS meta-analyses signals using gene expression and methylation *cis*-QTL data from primary cultures of foreskin melanocytes, isolated from foreskin of 106 newborn males^{54,76}. *Cis*-QTLs were assessed for variants in the ± 1 Mb region of each gene or CpG^{54,76}.”

Lines 597-600: “We tested all the significant eQTL genes or meQTL probes within ± 250 kb regions flanking the most significant GWAS SNP on each of the genome-wide regions of association (p -value $\leq 1.67e-08$) from the logistic meta-analyses summary statistics (11 different loci across the three GWAS models).”

We have also made it clearer on the legend of Table 1, which now reads as: “Colocalization results of expression and methylation *cis*-QTLs from cultured melanocytes (eQTL and meQTL, respectively) with GWAS SNPs on each hair colour category, showing colocalized SNPs with a posterior probability of ≥ 0.8 . We tested all the significant eQTL genes or meQTL probes within ± 250 kb regions flanking the GWAS lead

SNPs. The Gene/Methylation Annotation indicates the location of CpG probes with respect to the nearest gene, as well as relative to CpG island. NA = limited evidence of a single SNP driving the colocalization.”

Finally, we have also emphasized the approach used in the Results (lines 214-217): “We conducted colocalization analyses of the GWAS meta-analyses genome-wide signals using *hyprcoloc*⁵³ with gene expression and methylation *cis*-QTLs (eQTLs, meQTLs, respectively) to explore the putative regulatory role of the SNPs identified in our hair colour GWAS and identify candidate genes (Table 1 - See Methods for details).”

I'm also a bit confused by the comment that the *MC1R* variants wouldn't be eQTLs because they are non-synonymous; that might be the case here (but I want more information); but it isn't a general rule of eQTLs. Indeed *cis*-eQTLs are very much enriched for non-synonymous variants (see [dx.doi.org/10.1016/j.celrep.2017.05.018](https://doi.org/10.1016/j.celrep.2017.05.018)).

We agree with the reviewer in that non-synonymous variants may well be eQTLs in general. In the case of *MC1R* in particular, there is robust evidence indicating that the missense SNPs produce a loss-of-function of the protein which lead to red hair colour (and in some cases compound heterozygote missense variants also lead to red hair colour) (e.g. Rees, 2000), rather than through regulation of transcription of the gene. Overall, what we observe with the current datasets is that the missense variants do not colocalize with any eQTLs in the *MC1R* region (\pm 250 kb of the top SNP).

We have made this clearer in our Results (lines 229-232): "We did not find colocalization of QTLs on or near the gene *MC1R* for red hair colour. Given the current evidence, this is likely explained by the fact that known loss-of-function polymorphisms within *MC1R* lead to red hair colour, therefore they have a direct functional role on the translated protein. However, there is a possibility that we might be missing eQTLs beyond the 500 kb tested region."